# As Simple as Fine-tuning: LLM Alignment via Bidirectional Negative Feedback Loss

**Xin Mao**[1], **Huimin Xu**[1], **Feng-Lin Li**[2], **Ziqi Jin**[1], **Wang Chen**[2], **Wei Zhang**[3], **Anh Tuan Luu**[1]*
[1]Nanyang Technological University, [2]Shopee Pte. Ltd, [3]SEA Group
{xin.mao, anhtuan.luu, huimin.xu}@ntu.edu.sg, ZIQI007@e.ntu.edu.sg
{fenglin.li, chen.wang}@shopee.com, terry.zhang@sea.com

## Abstract

Direct Preference Optimization (DPO) has emerged as a more computationally efficient alternative to Reinforcement Learning from Human Feedback (RLHF) with Proximal Policy Optimization (PPO), eliminating the need for reward models and online sampling. Despite these benefits, DPO and its variants remain sensitive to hyper-parameters and prone to instability, particularly on mathematical datasets. We argue that these issues arise from the unidirectional likelihood-derivative negative feedback inherent in the log-likelihood loss function. To address this, we propose a novel LLM alignment loss that establishes a stable Bidirectional Negative Feedback (BNF) during optimization. Our proposed BNF loss eliminates the need for pairwise contrastive losses and does not require any extra tunable hyper-parameters or pairwise preference data, streamlining the alignment pipeline to be as simple as supervised fine-tuning. We conduct extensive experiments across two challenging QA benchmarks and four reasoning benchmarks. The experimental results show that BNF achieves comparable performance to the best methods on QA benchmarks, while its performance decrease on the four reasoning benchmarks is significantly lower compared to the best methods, thus striking a better balance between value alignment and reasoning ability. In addition, we further validate the performance of BNF on non-pairwise datasets, and conduct in-depth analysis of log-likelihood and logit shifts across different preference optimization methods. Github Url: https://github.com/MaoXinn/BNF.

## 1 Introduction

Alignment of Large Language Models (LLMs) plays a pivotal role in ensuring that these LLMs behave in accordance with human values and expectations (Bai et al., 2022b). As LLMs become increasingly integrated into various applications (Zhang et al., 2023; Roziere et al., 2023), ensuring proper alignment is crucial to mitigate risks such as biased or harmful outputs, while also enhancing trustworthiness. One of the most prominent LLM alignment methods is Reinforcement Learning from Human Feedback (Ouyang et al., 2022) with Proximal Policy Optimization (RLHF-PPO) (Schulman et al., 2017), which underpins the success of ChatGPT. However, despite its achievements, RLHF-PPO faces notable limitations, particularly regarding the high computational costs associated with reward modeling and online sampling (Casper et al., 2023). These challenges complicate its widespread adoption, especially in scenarios where computational resources are limited.

In response to these limitations, Direct Preference Optimization (DPO) (Rafailov et al., 2023) and its derivative methods (Ethayarajh et al., 2024; Zhao et al., 2023) aim to simplify the overall alignment pipeline by eliminating the need for reward and value models, as well as online sampling. Despite achieving impressive performance on QA and Chatbot tasks (Meng et al., 2024), DPO-series methods remain highly sensitive to hyper-parameters and often exhibit instability (Xu et al., 2024b). This instability is particularly pronounced when applied to mathematical datasets, leading to potential training collapse (Pal et al., 2024). Recently, several efforts (Xu et al., 2024a; Pal et al., 2024) have been made to address this issue. They attribute the collapse to an erroneous decrease in the likelihood of the preferred samples and propose using Negative Log Likelihood (NLL) regularization to stabilize the training process. Although these methods successfully prevent the model from collapsing on mathematical datasets, they perform poorly on several popular Chat and QA benchmarks (Meng et al., 2024) and introduce additional hyper-parameters.

Different from previous studies, we hypothesize that the instability of DPO may stem from a fundamental cause: the unidirectional likelihood-derivative negative feedback inherent in the log-likelihood loss. As shown in Figure 1, when applying the NLL loss to increase the likelihood $\pi_\theta(y_w|x) = \frac{e^{z_{y_w}}}{\sum_{k=1}^{|\mathcal{V}|} e^{z_k}}$

of a preferred output $y_w$, the partial derivative $\left|\frac{\partial \mathcal{L}_{\text{NLL}}}{\partial z_{y_w}}\right|$ with respect to the unnormalized logit $z_{y_w}$ will gradually decrease, which limits the rate of future increases in $z_{y_w}$ and $\pi_\theta(y_w|x)$, preventing the model from over-fitting. However, when the same loss is used to decrease the likelihood of a dispreferred output $y_l$, this negative feedback turns into a positive one. As $\pi_\theta(y_l|x)$ decreases, $\left|\frac{\partial -\mathcal{L}_{\text{NLL}}}{\partial z_{y_l}}\right|$ continues to rise, which further accelerates subsequent decreases in $z_{y_l}$ and $\pi_\theta(y_l|x)$, ultimately resulting in model collapse. To mitigate this issue,

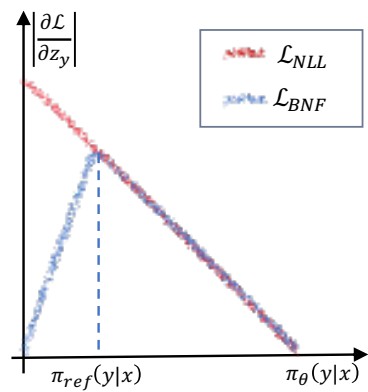

Figure 1: The likelihood-derivative curve of NLL and BNF loss.

DPO-series methods introduce pairwise contrastive losses to constrain the likelihood of $y_l$ from deviating excessively from that of the preferred sample $y_w$. However, this constraint is sensitive to hyper-parameters and lacks stability, resulting in the failure of DPO on mathematical datasets.

Based on the above findings, we propose a novel alignment loss, **B**idirectional **N**egative **F**eedback (BNF) loss. As shown in Figure 1, when optimizing with BNF loss, the partial derivative $\left|\frac{\partial \mathcal{L}}{\partial z_y}\right|$ reaches its maximum only when $\pi_\theta(y|x) = \pi_{\text{ref}}(y|x)$ (i.e., the initial state). Whether the likelihood $\pi_\theta(y|x)$ increases or decreases, $\left|\frac{\partial \mathcal{L}}{\partial z_y}\right|$ decreases linearly in both directions, thus establishing a bidirectional negative feedback. Since this bidirectional negative feedback fundamentally addresses the issue of excessive decreases in the likelihood of dispreferred samples, BNF eliminates the need for pairwise contrastive losses, further streamlining the alignment pipeline to be as simple as supervised fine-tuning. In summary, BNF loss offers the following advantages:

- **Less alignment tax**: Recent studies (Ouyang et al., 2022; Lin et al., 2023) have shown that using preference optimization methods to align LLMs with human values often harms their reasoning ability, referred to as the *alignment tax*. With the bidirectional negative feedback, our proposed BNF can naturally prevent the model from over-fitting to preference data, striking a better balance between human values and reasoning ability while minimizing the alignment tax.

- **Fewer tunable hyper-parameters**: Since the need for pairwise contrastive losses is eliminated, our proposed BNF loss does not involve any extra tunable hyper-parameters such as scaling factor $\beta$ in DPO (Rafailov et al., 2023), margin $\gamma$ in SimPO (Meng et al., 2024) or NLL regularization weight $\lambda$ in SLiC (Zhao et al., 2023), which significantly reduces the cost of grid searches.

- **Less pairwise data**: In addition to fewer tunable hyper-parameters, the removal of pairwise contrastive losses means that BNF no longer relies on pairwise preference data. Similar to KTO (Ethayarajh et al., 2024), BNF loss only requires either single preferred or dispreferred samples without the need for pairwise matching, significantly reducing data construction costs.

To comprehensively evaluate the effectiveness of our proposed BNF, we conduct extensive comparison experiments across two popular QA benchmarks and four reasoning benchmarks, using three 7B-9B LLMs as the base models. On two QA benchmarks, BNF achieves comparable performance to the best baselines, SimPO and DPO. Notably, we produce a top-performing model, based on Gemma-2-9B-it (Team et al., 2024), which outperforms not only all baselines but also larger scale LLMs like Gemma-2-27B-it and Nemotron-4-340B-instruct (Adler et al., 2024) on the most challenging benchmark, Wild-Bench (Lin et al., 2024). As for the reasoning benchmarks, BNF's perfor-

Table 1: Elo Rank on Wild-Bench.

| Model | Elo |
|---|---|
| GPT-4o (05-13) | 1237 |
| Claude-3-Opus | 1216 |
| Gemma-2-9B-BNF | 1186 |
| Nemotron-4-340B-it | 1184 |
| Gemma-2-27B-it | 1183 |
| Gemma-2-9B-SimPO | 1181 |
| Gemma-2-9B-DPO | 1181 |

mance decrease is significantly lower than that of SimPO (Meng et al., 2024) and DPO, indicating that BNF can strike a better balance between value alignment and reasoning abilities, thus paying less alignment tax. In addition, we design an experiment to validate the performance of BNF on non-pairwise datasets and conduct in-depth analysis of log-likelihood and logit shifts across different preference optimization methods.

## 2 THEORETICAL ANALYSIS OF LOG-LIKELIHOOD AND DPO

In this section, we first provide a detailed discussion on how the unidirectional negative feedback leads to an excessive decrease in the likelihood of dispreferred samples, ultimately causing model collapse. Then, we explain how DPO-series methods mitigate this excessive decrease by employing a pairwise contrastive loss, and why this approach is ineffective for mathematical datasets.

### 2.1 LIMITATION OF LOG-LIKELIHOOD LOSS

In the Supervised Fine-Tuning (SFT) stage, we typically only use the NLL loss to maximize the likelihood of each sample from the dataset without worrying about collapse. In the alignment stage, a naive approach would be to try the following log-likelihood loss for preference optimization:

$$\mathcal{L}_{\text{NLL+PLL}} = \mathbb{E}_{(x,y_w,y_l)\sim\mathcal{D}}[-\log \pi_\theta(y_w|x) + \log \pi_\theta(y_l|x)] \tag{1}$$

This naive loss tries to increase the likelihood of the preferred sample $y_w$ with NLL and decrease the likelihood of the dispreferred sample $y_l$ with Positive Log Likelihood (PLL). However, LLMs will quickly collapse when optimized with the above loss (Rafailov et al., 2023). As mentioned in Section 1, the underlying reason is the unidirectional likelihood-derivative negative feedback inherent in the log-likelihood loss. Let's consider a simple case where the response $y$ consists of only a single token. When we apply the NLL loss to increase the likelihood $\pi_\theta(y_w|x)$ of the preferred $y_w$:

$$\mathcal{L}_{\text{NLL}} = \mathbb{E}_{(x,y_w)\sim\mathcal{D}}[-\log \pi_\theta(y_w|x)] = \mathbb{E}_{(x,y_w)\sim\mathcal{D}}\left[-\log \frac{e^{z_{y_w}}}{\sum_{k=1}^{|\mathcal{V}|} e^{z_k}}\right] \tag{2}$$

where $|\mathcal{V}|$ is the vocabulary size. The partial derivative of the NLL loss with respect to the logit $z_{y_w}$ is given by $\left|\frac{\partial \mathcal{L}_{\text{NLL}}}{\partial z_{y_w}}\right| = 1 - \pi_\theta(y_w|x)$ (derivation in Appendix A.1). In this case, the likelihood $\pi_\theta(y_w|x)$ and the partial derivative $\left|\frac{\partial \mathcal{L}_{\text{NLL}}}{\partial z_{y_w}}\right|$ actually establish a stable negative feedback. As $\pi_\theta(y_w|x)$ increases, $|\frac{\partial \mathcal{L}_{\text{NLL}}}{\partial z_{y_w}}|$ gradually decreases, which limits the rate of future increases in $z_{y_w}$ and $\pi_\theta(y_w|x)$, thereby preventing the model from over-fitting.

However, this negative feedback is unidirectional. When $\mathcal{L}_{\text{PLL}} = \mathbb{E}_{(x,y_l)\sim\mathcal{D}}[\log \pi_\theta(y_l|x)]$ is used to decrease the likelihood $\pi_\theta(y_l|x)$, the above negative feedback will turn into a positive one, which means that any decrease in $\pi_\theta(y_l|x)$ will further accelerate itself. As $\pi_\theta(y_l|x)$ decreases, the partial derivative $|\frac{\partial \mathcal{L}_{\text{PLL}}}{\partial z_{y_l}}| = 1 - \pi_\theta(y_l|x)$ continues to rise, which further accelerates subsequent decreases in $z_{y_l}$ and $\pi_\theta(y_l|x)$, ultimately resulting in model collapse. Similarly, since the log-likelihood of a longer response is the sum of the log-likelihoods of individual tokens $\log \pi_\theta(y|x) = \sum_i \log \pi_\theta(y_i|x, y_{<i})$, the aforementioned conclusion still holds.

### 2.2 ROLE OF PAIRWISE CONTRASTIVE LOSSES

Since solely using vanilla log-likelihood loss in preference optimization will cause model collapse, DPO-series methods introduce pairwise contrastive losses to stabilize the optimization process. The gradients of most DPO-series methods with respect to parameters $\theta$ could be written as follows:

$$\nabla_\theta \mathcal{L} = \mathbb{E}_{(x,y_w,y_l)\sim\mathcal{D}}\left[\underbrace{\mathcal{C}(y_l, y_w, \pi_\theta, \pi_{\text{ref}})}_{\text{constrain the log-likelihoods gap}}\left[-\underbrace{\nabla_\theta \log \pi_\theta(y_w|x)}_{\text{increase likelihood of } y_w} + \underbrace{\nabla_\theta \log \pi_\theta(y_l|x)}_{\text{decrease likelihood of } y_l}\right]\right] \tag{3}$$

The second half of Eq. (3) is similar to the gradient of Eq. (1), which aims to increase the likelihood of the preferred sample $y_w$ and decrease the likelihood of the dispreferred sample $y_l$. The key difference is the introduction of a pairwise contrastive function $\mathcal{C}(y_l, y_w, \pi_\theta, \pi_{\text{ref}})$, which constrains the likelihood of dispreferred samples from deviating excessively from that of the preferred ones. Taking DPO as an example, $\mathcal{C}_{\text{DPO}}(y_l, y_w, \pi_\theta, \pi_{\text{ref}}) = \beta \cdot \sigma\left(\beta \log \frac{\pi_\theta(y_l|x)}{\pi_{\text{ref}}(y_l|x)} - \beta \log \frac{\pi_\theta(y_w|x)}{\pi_{\text{ref}}(y_w|x)}\right)$, where $\mathcal{C}_{\text{DPO}}$ decreases gradually as the log-likelihood gap between $y_l$ and $y_w$ increases, creating a negative feedback that constrains $\log \pi_\theta(y_l|x)$ from deviating excessively from $\log \pi_\theta(y_w|x)$ . Table 2 lists several representative DPO-series methods and their derived constraint functions $\mathcal{C}$. Despite variations in specific implementations, these methods share a common core idea: using pairwise contrastive losses to avoid excessive decreases in the likelihood of dispreferred samples.

Table 2: The objective $\mathcal{L}$ and derived constraint $\mathcal{C}$ of representative DPO-series methods.

| Method | Objective $\mathcal{L}$ | $\mathcal{C}(y_w, y_l, \pi_\theta, \pi_{\text{ref}})$ |
|---|---|---|
| SLiC-HF (Zhao et al., 2023) | $\max\left(0, \delta - \log \pi_\theta(y_w\|x) + \log \pi_\theta(y_l\|x)\right)$ | $f(x) = \begin{cases} 1, & \text{if } \log \pi_\theta(y_w\|x) - \log \pi_\theta(y_l\|x) < \delta \\ 0, & \text{if } \log \pi_\theta(y_w\|x) - \log \pi_\theta(y_l\|x) >= \delta \end{cases}$ |
| DPO (Rafailov et al., 2023) | $-\log \sigma\left(\beta \log \frac{\pi_\theta(y_w\|x)}{\pi_{\text{ref}}(y_w\|x)} - \beta \log \frac{\pi_\theta(y_l\|x)}{\pi_{\text{ref}}(y_l\|x)}\right)$ | $\beta \cdot \sigma\left(\beta \log \frac{\pi_\theta(y_l\|x)}{\pi_{\text{ref}}(y_l\|x)} - \beta \log \frac{\pi_\theta(y_w\|x)}{\pi_{\text{ref}}(y_w\|x)}\right)$ |
| IPO (Azar et al., 2023) | $\left(\log \frac{\pi_\theta(y_w\|x)}{\pi_{\text{ref}}(y_w\|x)} - \log \frac{\pi_\theta(y_l\|x)}{\pi_{\text{ref}}(y_l\|x)} - \frac{1}{2\tau}\right)^2$ | $2 \cdot \left(\frac{1}{2\tau} + \log \frac{\pi_\theta(y_l\|x)}{\pi_{\text{ref}}(y_l\|x)} - \log \frac{\pi_\theta(y_w\|x)}{\pi_{\text{ref}}(y_w\|x)}\right)$ |
| SimPO (Meng et al., 2024) | $-\log \sigma\left(\frac{\beta}{\|y_w\|} \log \pi_\theta(y_w\|x) - \frac{\beta}{\|y_l\|} \log \pi_\theta(y_l\|x) - \gamma\right)$ | $\sigma\left(\gamma + \frac{\beta}{\|y_l\|} \log \pi_\theta(y_l\|x) - \frac{\beta}{\|y_w\|} \log \pi_\theta(y_w\|x)\right)$ |

## 2.3 FAILURE ON MATHEMATICAL DATASETS

Generally, constructing a preference dataset involves three steps: (1) generating multiple responses for each prompt in the instruction dataset, (2) scoring each response with human annotators or reward models, and (3) selecting the highest and lowest scoring responses as the preference pair. However, since mathematical reasoning requires rigorous logic, the diversity of responses to math problems is often significantly lower compared to other types of problems. As a result, mathematical preference datasets often contain many highly similar pairs, some differing by even only one single token [1]. Given this data distribution, selecting an appropriate scaling factor $\beta$ for DPO-series methods is challenging, as it is difficult to create a substantial log-likelihood gap for highly similar preference pairs. As illustrated in Figure 2, although the likelihood of the correct answer is significantly greater than that of the incorrect one (0.8 vs. 0.05), the total log-likelihood gap still only amounts to 1.2. In this situation, if the scaling factor $\beta$ is set as usually (e.g., 0.1 in DPO), the scaled gap will become extremely small, which may cause the pairwise contrastive function $\mathcal{C}$ to fail in preventing the excessive decrease of $\pi_\theta(y_l|x)$. Conversely, if $\beta$ is set higher than usual, it may lead to underfitting for the preference pairs with less overlap, which also negatively affects performance. Recent studies have proposed two solutions: (1) adding NLL regularization to maximize the likelihood of preferred responses (Saeidi et al., 2024), and (2) directly removing preference pairs with small edit distances from the dataset (Pal et al., 2024). Different from the above methods, this paper addresses this issue through a more fundamental approach, introducing bidirectional negative feedback, which enables preference optimization to eliminate reliance on contrastive loss functions and remove the need for tuning any hyper-parameters. In Appendix D, we explore the performance of different alignment methods on a purely mathematical preference dataset to further validate the effectiveness.

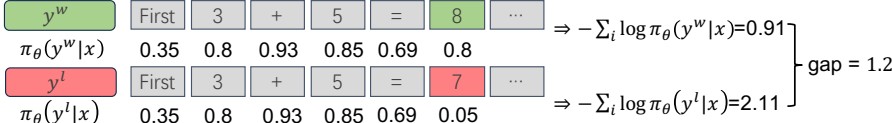

Figure 2: An example of mathematical preference dataset. Due to the significant overlap between preferred and dispreferred samples, it is difficult to create a substantial log-likelihood gap.

## 3 BIDIRECTIONAL NEGATIVE FEEDBACK LOSS

### 3.1 OPTIMIZATION OBJECTIVE

In Section 2, we argue that the limitation of the log-likelihood loss in preference optimization stems from the unidirectional negative feedback. While DPO-series methods introduce pairwise contrastive losses to address this issue, they face challenges in hyper-parameter tuning, especially when applied to mathematical preference datasets. Building on these insights, we propose a novel alignment loss that establishes a bidirectional negative feedback, eliminating the need for contrastive losses and simplifying LLM alignment to the level of supervised fine-tuning. Given a policy model $\pi_\theta$, a reference model $\pi_{\text{ref}}$, and a preference dataset $\mathcal{D}$ where each response is labeled as either preferred or not (without necessarily being paired), the optimization objective of our proposed BNF

---

[1]https://huggingface.co/datasets/argilla/distilabel-math-preference-dpo

loss is described as follows:

$$\mathcal{L}_{\text{BNF}} = -\mathbb{E}_{(x,y)\sim\mathcal{D}} \left[ \frac{\text{label}(y)}{|y|} \sum_{i}^{|y|} \sum_{j}^{|\mathcal{V}|} f_{\text{BNF}}(y_i, t_j) \log \pi_\theta(t_j|x, y_{<i}) \right] \tag{4}$$

where $|y|$ represents the length of the response $y$, $\text{label}(y) = \begin{cases} 1, & \text{if } y \text{ is preferred} \\ -1, & \text{if } y \text{ is dispreferred} \end{cases}$ is the annotation label of $y$, $|\mathcal{V}|$ is the vocabulary size, and $t_j$ is the $j$-th token in the vocabulary. In fact, Eq (4) is merely a standard length-normalized cross-entropy loss. If we adopt the following one-hot $f_{\text{BNF}} = f_{\text{LL}}$ as the target distribution, the proposed loss will be same with Eq. (1):

$$f_{\text{LL}}(y_i, t_j) = \begin{cases} 1, & \text{if } y_i = t_j \\ 0, & \text{if } y_i \neq t_j \end{cases} \tag{5}$$

Therefore, the core of our proposed method lies in the target distribution $f_{\text{BNF}}(y_i, t_j)$ of the cross-entropy loss, which we call Dynamic Target Distribution (DTD):

$$f_{\text{BNF}}(y_i, t_j) = \begin{cases} sg\left[\min\left(\frac{\pi_\theta(y_i|x, y_{<i})}{\pi_{\text{ref}}(y_i|x, y_{<i})}, 1\right)\right], & \text{if } y_i = t_j \\ sg\left[\frac{1 - f_{\text{BNF}}(y_i, y_i)}{1 - \pi_\theta(y_i|x, y_{<i})}\pi_\theta(t_j|x, y_{<i})\right], & \text{if } y_i \neq t_j \end{cases} \tag{6}$$

where $sg$ represents the stop gradient operation (i.e., *detach* in PyTorch). Since $f_{\text{BNF}}(y_i)$ satisfies $f_{\text{BNF}}(y_i, t_j) \geq 0$ and $\sum_j^{|\mathcal{V}|} f_{\text{BNF}}(y_i, t_j) = 1$ for all $y_i$ and $t_j$ (derivation in Appendix A.2), it constitutes a valid probability distribution. Although DTD seems complex at first glance, its code implementation is quite simple and efficient (as shown in Appendix B). Moreover, compared to DPO-series methods, the BNF loss involves no tunable hyper-parameters and eliminates the need for pairwise preference data, which reduces the costs of grid searches and dataset construction. In the next section, we will explain how this loss function establishes a bidirectional negative feedback.

## 3.2 GRADIENT ANALYSIS

For a mechanistic understanding of the proposed BNF loss, we need to analyze the gradient of the loss function $\mathcal{L}_{\text{BNF}}$. The gradient with respect to parameters $\theta$ can be written as:

$$\nabla_\theta \mathcal{L}_{\text{BNF}} = -\mathbb{E}_{(x,y)\sim\mathcal{D}} \sum_{i}^{|y|} \sum_{j}^{|\mathcal{V}|} \frac{\partial \mathcal{L}_{\text{BNF}}}{\partial z_{t_j}^{x, y_{<i}}} \nabla_\theta z_{t_j}^{x, y_{<i}} \tag{7}$$

where $z_{t_j}^{x, y_{<i}}$ represents the original ouput logit of $\pi_\theta(t_j|x, y_{<i})$ before Softmax . Since $f_{\text{BNF}}(y_i)$ is a valid probability distribution with stop gradient, the partial derivative $\frac{\partial \mathcal{L}_{\text{BNF}}}{\partial z_{t_k}^{x, y_{<i}}}$ with respect to any output logits $z_{t_k}^{x, y_{<i}}$ can be obtained as follows (derivation in Appendix A.3):

$$\frac{\partial \mathcal{L}_{\text{BNF}}}{\partial z_{t_k}^{x, y_{<i}}} = \frac{\text{label}(y)}{|y|}\left[\pi_\theta(t_k|x, y_{<i}) - f_{\text{BNF}}(y_i, t_k)\right] \tag{8}$$

By substituting Eq. (6) into Eq. (8), we can derive a function between $\left|\frac{\partial \mathcal{L}_{\text{BNF}}}{\partial z_{t_k}^{x, y_{<i}}}\right|$ and $\pi_\theta(t_k|x, y_{<i})$ (as shown in Appendix A.4). Here, we focus on the token $t_k = y_i$ within the responses $y$:

$$\left|\frac{\partial \mathcal{L}_{\text{BNF}}}{\partial z_{y_i}^{x, y_{<i}}}\right| = \frac{\text{label}(y)}{|y|} \cdot \begin{cases} \pi_\theta(y_i|x, y_{<i})\frac{1 - \pi_{\text{ref}}(y_i|x, y_{<i})}{\pi_{\text{ref}}(y_i|x, y_{<i})}, & \text{if } \pi_\theta(y_i|x, y_{<i}) < \pi_{\text{ref}}(y_i|x, y_{<i}) \\ 1 - \pi_\theta(y_i|x, y_{<i}), & \text{if } \pi_\theta(y_i|x, y_{<i}) \geq \pi_{\text{ref}}(y_i|x, y_{<i}) \end{cases} \tag{9}$$

From this piecewise function, we can observe that the partial derivative $\left|\frac{\partial \mathcal{L}_{\text{BNF}}}{\partial z_{y_i}^{x, y_{<i}}}\right|$ reaches its maximum only when $\pi_\theta(y_i|x, y_{<i}) = \pi_{\text{ref}}(y_i|x, y_{<i})$. When $\pi_\theta(y_i|x, y_{<i}) < \pi_{\text{ref}}(y_i|x, y_{<i})$, since $\frac{1 - \pi_{\text{ref}}(y_i|x, y_{<i})}{\pi_{\text{ref}}(y_i|x, y_{<i})}$ is always greater than 0, $\left|\frac{\partial \mathcal{L}_{\text{BNF}}}{\partial z_{y_i}^{x, y_{<i}}}\right|$ decreases linearly as $\pi_\theta(y_i|x, y_{<i})$ decreases until it reaches 0. In contrast, when $\pi_\theta(y_i|x, y_{<i}) \geq \pi_{\text{ref}}(y_i|x, y_{<i})$, $\left|\frac{\partial \mathcal{L}_{\text{BNF}}}{\partial z_{y_i}^{x, y_{<i}}}\right|$ is equal to that of the NLL loss, decreasing linearly as $\pi_\theta(y_i|x, y_{<i})$ increases. In this way, the proposed BNF loss establishes a bidirectional negative feedback between the partial derivatives and the likelihoods, aligning precisely with the illustration in Figure 1. In Appendix C, we will give more discussions in detail.

Table 3: Statistical comparison of two instruction-following QA benchmarks.

| Dataset | #Tasks | #Turns | ChatHistory | QueryLen | PromptLen | RealUser | Evaluation |
|---------|--------|--------|-------------|----------|-----------|----------|------------|
| **ArenaHard** | 500 | 1 | ✗ | 406 | 406 | ✓ | Pair |
| **Wild-Bench** | 1,024 | ≤5 | ✓ | 979 | 3,402 | ✓ | Score+Pair |

## 4 EXPERIMENTAL SETUP

We follow the experimental setup of SimPO (Meng et al., 2024) to objectively assess the effectiveness of our proposed method. They provide numerous checkpoints aligned with DPO-series methods and the corresponding training datasets, we acknowledge their contributions. For a reference, we list the GPU hours for training and API cost for evaluation in Appendix F.

**Models and training datasets.** In this paper, we adopt three mainstream open-source LLMs as the base models: Mistral-7B-Instruct-v0.2 (Jiang et al., 2023a), Meta-Llama-3-8B-Instruct (Dubey et al., 2024), and Gemma-2-9b-it (Team et al., 2024). For a fair comparison, we use the same preference training datasets constructed by SimPO. Specifically, for each prompt $x$ in Ultrafeedback (Cui et al., 2024), they generate 5 responses with a sampling temperature of 0.8. Then, using PairRM (Jiang et al., 2023b) or ArmoRM (Wang et al., 2024a) to score the 5 responses, selecting the highest-scoring one as $y_w$ and the lowest-scoring one as $y_l$. Here are the three training datasets: Mistral-Ultrafeedback-PairRM, Llama3-Ultrafeedback-ArmoRM, and Gemma2-Ultrafeedback-ArmoRM.

**Training hyper-parameters.** In this paper, we set the maximum sequence length to 4096 and adopt the AdamW optimizer (Loshchilov & Hutter, 2018), applying cosine learning rate schedule with 10% warm-up steps. Since our proposed BNF loss does not have any extra tunable hyper-parameters, we only perform grid searches on batch size {64, 128, 256} and learning rate {5e-7, 6e-7, 8e-7, 1e-6}. After grid search, we adopt a unified batch size of 128 and select learning rates of 5e-7 for Mistral-7B-Instruct-v0.2, 6e-7 for Meta-Llama-3-8B-Instruct, and 8e-7 for Gemma-2-9b-it.

**Baselines.** In addition to DPO (Rafailov et al., 2023), we also compare our proposed BNF loss with the following strong baselines in preference optimization: (1) SLiC-HF (Zhao et al., 2023) is based on contrastive ranking loss and integrates an NLL loss as the regularization term. (2) IPO (Azar et al., 2023) is a theoretically grounded method designed to bypass DPO's assumption that pairwise preferences can be replaced by pointwise rewards. (3) KTO (Ethayarajh et al., 2024) learns from preference data that is not pairwise. (4) CPO (Xu et al., 2024a) introduces an NLL regularization term to the DPO loss function. (5) SimPO (Meng et al., 2024) is a simpler and effective preference optimization method without using a reference model. All the baselines have been well-tuned through hyper-parameter grid searches, as described in Meng et al. (2024).

**Evaluation benchmarks.** We primarily evaluate all the models using two recent proposed instruction-following QA benchmarks: Arena-Hard (Li et al., 2024) and Wild-Bench(Lin et al., 2024). Arena-Hard, an enhanced version of MT-Bench (Zheng et al., 2023), includes 500 high-quality prompts from real user queries. For Arena-Hard, we report the standard win rate (WR) and length-controlled win rate (LC), using GPT-4-0314 as the reference model and GPT-4o-mini as the evaluator[2]. Wild-Bench is a highly challenging benchmark, featuring longer prompts and more difficult questions sourced from real users. Besides scoring each response on a 100-point scale, Wild-Bench further introduces LMSYS-Elo (Chiang et al., 2024) to better rank all the models. For Wild-Bench, we report Elo and score using GPT-4o as the evaluator. The statistical comparison of these benchmarks are listed in Table 3. The reason we do not adopt MT-bench and AlpacaEval (Dubois et al., 2024) is due to the significant flaws in these datasets: MT-bench only contains 80 samples and exhibits poor separability across different methods (Meng et al., 2024). AlpacaEval is a highly imbalanced dataset, with 50% of the question types focused on information seeking, while less than 20% are related to reasoning. In fact, Arena-Hard and Wild-Bench are upgraded versions of MT-Bench and AlpacaEval, offering more challenging tasks with a more balanced distribution. Furthermore, we also evaluate all the models on four logical reasoning benchmarks to verify the impact of these alignment methods on model's reasoning abilities, including: MMLU-redux (Gema et al., 2024) (Language), CRUX (Gu et al., 2024) (Code), GSM8K (Cobbe et al., 2021) and MATH-L5 (Hendrycks et al., 2021) (Math). For these reasoning benchmarks, we use ZeroEval (Lin, 2024) as the evaluator, which aims to evaluate instruction-tuned LLMs for their zero-shot performance.

---

[2]The original evaluator was GPT-4-Turbo, we replace it with GPT-4o-mini due to the high API costs.

Table 4: Main experimental results across all benchmarks. The numbers in parentheses ($x$) indicate the relative ranking of each method under this metric, while Average Rank represents the average ranking across all metrics for each method. For Gemma-2, since Meng et al. (2024) only provides the checkpoints for DPO and SimPO, BNF is compared exclusively with these two methods.

| Method | Meta-Llama-3-8B-Instruct | | | | | | | |
|---|---|---|---|---|---|---|---|---|
| | Wild-Bench | | Arena-Hard | | GSM8K | MATH | CRUX | MMLU | Average |
| | Elo | Score | LC (%) | WR(%) | Acc (%) | Acc (%) | Acc (%) | Acc (%) | Rank |
| Base | 1131 (8) | 29.2 (8) | 33.9 (8) | 36.1 (8) | 78.5 (4) | 6.8 (7) | 38.1 (4) | 62.4 (3) | 6.25 |
| SLiC-HF | 1142 (5) | 33.2 (7) | 49.1 (5) | 46.3 (5) | 66.6 (7) | 8.3 (6) | 33.9 (6) | 61.4 (4) | 5.63 |
| DPO | 1155 (1) | 39.5 (1) | 60.4 (1) | 62.2 (1) | 70.5 (6) | 8.5 (5) | 31.8 (8) | 56.4 (7) | 3.75 |
| IPO | 1146 (4) | 36.1 (4) | 50.1 (4) | 51.3 (4) | 79.9 (1) | 9.2 (4) | 39.2 (1) | 59.8 (6) | 3.50 |
| KTO | 1139 (6) | 34.7 (5) | 45.4 (6) | 45.8 (6) | 78.8 (3) | 10.0 (1) | 38.9 (2) | 63.7 (1) | 3.75 |
| CPO | 1136 (7) | 34.1 (6) | 44.6 (7) | 44.9 (7) | 79.2 (2) | 9.3 (2) | 38.7 (3) | 63.5 (2) | 4.50 |
| SimPO | 1149 (3) | 37.2 (3) | 52.2 (3) | 52.5 (2) | 56.6 (8) | 6.0 (8) | 32.6 (7) | 55.0 (8) | 5.25 |
| BNF | 1153 (2) | 37.5 (2) | 54.8 (2) | 52.1 (3) | 77.0 (5) | 9.3 (2) | 35.4 (5) | 61.4 (4) | **3.13** |

| Method | Mistral-7B-Instruct-v0.2 | | | | | | | |
|---|---|---|---|---|---|---|---|---|
| | Wild-Bench | | Arena-Hard | | GSM8K | MATH | CRUX | MMLU | Average |
| | Elo | Score | LC (%) | WR(%) | Acc (%) | Acc (%) | Acc (%) | Acc (%) | Rank |
| Base | 1098 (8) | 25.6 (8) | 19.2 (8) | 18.9 (8) | 42.7 (2) | 3.9 (1) | 25.1 (3) | 53.0 (3) | 5.13 |
| SLiC-HF | 1130 (3) | 31.2 (2) | 28.9 (4) | 31.0 (3) | 43.6 (1) | 2.6 (6) | 24.3 (4) | 52.1 (6) | 3.63 |
| DPO | 1127 (4) | 29.7 (6) | 29.8 (3) | 30.4 (4) | 42.2 (5) | 3.1 (3) | 22.3 (7) | 53.2 (2) | 4.25 |
| IPO | 1121 (7) | 26.9 (7) | 25.4 (7) | 26.1 (7) | 35.8 (7) | 3.6 (2) | 28.3 (1) | 50.7 (8) | 5.75 |
| KTO | 1126 (5) | 30.2 (4) | 27.8 (6) | 27.7 (6) | 42.6 (3) | 2.9 (5) | 23.3 (6) | 52.8 (4) | 4.75 |
| CPO | 1124 (6) | 29.9 (5) | 28.1 (5) | 28.3 (5) | 42.3 (4) | 3.0 (4) | 25.3 (2) | 51.9 (7) | 4.88 |
| SimPO | 1133 (1) | 32.0 (1) | 36.0 (1) | 36.6 (1) | 33.7 (8) | 2.2 (8) | 21.4 (8) | 52.2 (5) | 4.13 |
| BNF | 1131 (2) | 31.1 (3) | 35.2 (2) | 34.5 (2) | 39.7 (6) | 2.6 (6) | 24.3 (4) | 54.8 (1) | **3.25** |

| Method | Gemma-2-9b-It | | | | | | | |
|---|---|---|---|---|---|---|---|---|
| | Wild-Bench | | Arena-Hard | | GSM8K | MATH | CRUX | MMLU | Average |
| | Elo | Score | LC (%) | WR(%) | Acc (%) | Acc (%) | Acc (%) | Acc (%) | Rank |
| Base | 1160 (4) | 42.7 (4) | 54.9 (4) | 54.5 (4) | 87.9 (3) | 19.9 (3) | 44.6 (2) | 72.7 (2) | 3.25 |
| DPO | 1181 (2) | 53.2 (2) | 77.3 (2) | 81.3 (1) | 88.5 (1) | 20.2 (2) | 44.3 (3) | 72.8 (1) | 1.75 |
| SimPO | 1181 (2) | 53.3 (1) | 72.6 (3) | 75.4 (3) | 87.7 (4) | 18.2 (4) | 41.4 (4) | 72.6 (4) | 3.13 |
| BNF | 1186 (1) | 53.2(2) | 77.5 (1) | 80.8 (2) | 88.0 (2) | 21.8 (1) | 45.3 (1) | 72.7 (2) | **1.50** |

## 5 EXPERIMENTAL RESULTS

Table 4 presents the detailed experimental results of all preference optimization methods across six benchmarks and three base models, covering both performance and relative rankings. In this section, we first provide a comprehensive analysis of these statistics. Then, we conduct an experiment to evaluate the performance of our proposed BNF on non-pairwise preference datasets. Finally, we analyze the log-likelihood and logit shifts under different preference optimization methods. In addition, we provide some response comparisons in Appendix G for reference.

### 5.1 MAIN EXPERIMENT ANALYSIS

**Preference optimization is essential for QA.** From Table 4, we observe that all three base models rank the lowest on two QA benchmarks, Wild-Bench and Arena-Hard, while all preference optimization methods achieve significant performance improvements. More specifically, DPO, SimPO, and our proposed BNF demonstrate superior performance compared to other methods, consistently securing the top three positions across all QA metrics. DPO performs best on Meta-Llama-3, SimPO works best on Mistral-7B, and our proposed BNF exhibits outstanding performance on Gemma-2. These experimental results indicate that preference optimization is essential for improving model performance on QA tasks and demonstrate that our proposed BNF can deliver performance comparable to, or even superior to, the strongest preference optimization baselines.

**BNF pays the lowest alignment tax.** When our attention shifts to reasoning benchmarks, we find that the methods excelling in QA benchmarks often perform poorly in reasoning benchmarks, which is also referred to as alignment tax (Ouyang et al., 2022). For instance, SimPO, which performs well on QA, experiences a significant drop in performance on reasoning, with a more than 10% performance decrease on GSM8K, ranking nearly at the bottom. In contrast, KTO, which excels in reasoning benchmarks, performs poorly on QA benchmarks, only slightly better than base models. Some studies (Lin et al., 2023) suggest that alignment tax occurs due to over-fitting to preference data, leading to a decline in reasoning ability. With the bidirectional negative feedback, our proposed BNF can automatically constrain the model from over-fitting to preference datasets, thus preserving the reasoning ability. Experimental results show that BNF achieves the best average relative ranking across three base models (last column of Table 4), indicating that BNF strikes a better balance between QA and reasoning ability, thus paying the lowest alignment tax.

**Response length of different methods.** Some recent studies (Xu et al., 2024b) argue that the models trained with DPO tend to produce verbose responses that may degrade the quality of results. To investigate this issue, we calculate the average response length of all preference optimization methods on two QA benchmarks and list their statistics in Table 5. On Meta-Llama-3, the average response lengths of almost all methods decrease. However, on Mistral-7B, only KTO's response length decrease on both two benchmarks, while all other methods see an increase on at least one benchmark. Despite the introduction of length normalization, the response length of SimPO on Mistral-7B still

Table 5: Average output length.

| Method | Meta-Llama-3 | | Mistral-7B | |
|---|---|---|---|---|
| | **WB** | **AH** | **WB** | **AH** |
| **Base** | 2975 | 595 | 2832 | 507 |
| **SLiC-HF** | 2318 ↓ | 469 ↓ | 3198 ↑ | 576 ↑ |
| **DPO** | 2665 ↓ | 602 ↑ | 2655 ↓ | 509 ↑ |
| **IPO** | 2843 ↓ | 547 ↓ | 3431 ↑ | 545 ↑ |
| **KTO** | 2776 ↓ | 543 ↓ | 2813 ↓ | 504 ↓ |
| **ORPO** | 2551 ↓ | 523 ↓ | 2913 ↑ | 524 ↑ |
| **SimPO** | 2533 ↓ | 527 ↓ | 3214 ↑ | 521 ↑ |
| **BNF** | 2521 ↓ | 496 ↓ | 2827 ↓ | 526 ↑ |

increase significantly. These statistics indicate that average response lengths are closely related to the base models. DPO does not necessarily generate longer responses, and specially designed method may also fail on certain models.

## 5.2 NON-PAIRWISE OPTIMIZATION

With the bidirectional negative feedback, BNF no longer requires pairwise contrastive losses to constrain the excessive decrease in the likelihood of dispreferred samples. Therefore, BNF fundamentally eliminates the need for preference pairs during optimization. To evaluate its performance with non-pairwise preference datasets, we randomly mask either the preferred or dispreferred response from the original preference pairs with a certain probability. Table 6 presents the experimental results with different pairing ratios. On QA benchmarks, it is evident that more pairwise preference data leads to better performance. However, even without any preference pairs, BNF still achieves significant performance improvements over the base model, with an average score increase of 6.8 on Wild-Bench and a 12.3% win rate improvement on Arena-Hard. Interestingly, the alignment tax phenomenon is also observed in this scenario. At a 0% pairing ratio, the absence of preference pairs prevents over-fitting to preference data. While this leads to lower performance gains in QA, the model's reasoning ability improves on mathematical datasets, showing a 1.6% increase on GSM8K and a 2.6% improvement on Math-L5.

Table 6: Experimental results with different pairing ratios. A ratio of 50% indicates that in half of the preference pairs, one response is randomly masked. A ratio of 0% means that no pairwise data is included in the dataset, while 100% represents the original preference dataset.

| Pairing-Ratio | Meta-Llama-3-8B-Instruct | | | | | | | | |
|---|---|---|---|---|---|---|---|---|---|
| | **Wild-Bench** | | **Arena-Hard** | | **GSM8K** | **MATH** | **CRUX** | **MMLU** | **Average** |
| | **Elo** | **Score** | **LC** (%) | **WR**(%) | **Acc** (%) | **Acc** (%) | **Acc** | **Acc** (%) | **Rank** |
| **Base** | 1131 (5) | 29.2 (5) | 33.9 (5) | 36.1 (5) | 78.5 (2) | 6.8 (5) | 38.1 (1) | 62.4 (1) | 3.63 |
| **0%** | 1146 (4) | 36.0 (4) | 49.3 (4) | 48.4 (4) | 80.1 (1) | 9.4 (2) | 35.6 (4) | 62.2 (2) | 3.13 |
| **25%** | 1148 (2) | 36.2 (3) | 51.5 (3) | 49.7 (3) | 77.9 (3) | 9.3 (3) | 36.7 (3) | 61.8 (3) | 2.88 |
| **50%** | 1148 (2) | 36.7 (2) | 53.5 (2) | 51.4 (2) | 76.4 (5) | 10.0 (1) | 37.9 (2) | 61.2 (5) | 2.63 |
| **100%** | 1153 (1) | 37.5 (1) | 54.8 (1) | 52.1 (1) | 77.0 (4) | 9.3 (3) | 35.4 (5) | 61.4 (4) | 2.5 |

## 5.3 DISTRIBUTIONS OF LOG-LIKELIHOOD AND LOGIT

To gain a deeper understanding of the optimization behavior of BNF, we analyze log-likelihood and logit shifts between policy and reference models using $1,000$ questions from Wild-Bench (Lin et al., 2024). We apply greedy decoding to generate responses using BNF and three baselines: DPO, IPO, and SimPO. The base model is Llama-3-8B-Instruct.

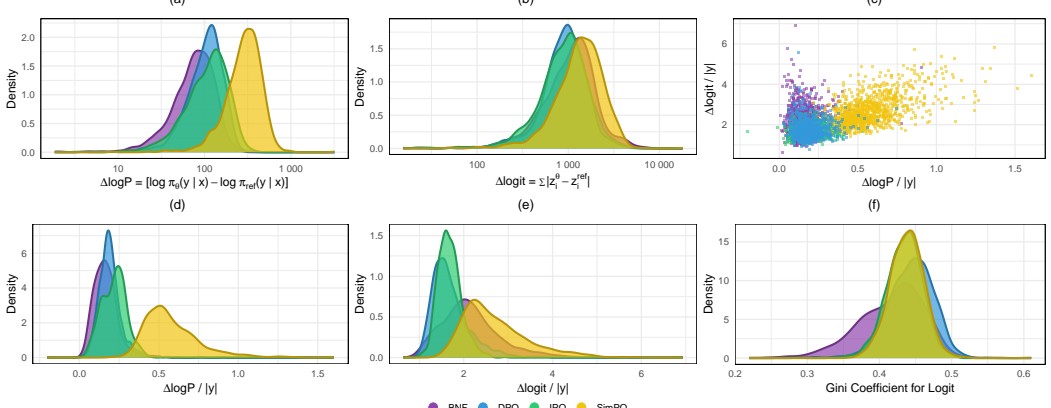

Figure 3: Comparisons between BNF, DPO, IPO, and SimPO. **(a)** Log-likelihood shifts. **(b)** Absolute logit shifts. **(c)** Logit shifts vs. log-likelihood shifts. **(d)** Length-normalized log-likelihood shifts. **(e)** Length-normalized absolute logit shifts. **(f)** Gini coefficients for logits.

**BNF exhibits minimal log-likelihood shift.** Our experiments reveal that BNF exhibits the least log-likelihood shift (Figure 3a and 3d), which may help preserve reasoning and comprehension capabilities from the reference model. In contrast, SimPO shows the largest shift, potentially explaining its inferior performance in these areas.

**BNF's uniform logit increase leads to unique shift pattern.** Interestingly, BNF shows larger logit shifts compared to DPO and IPO (Figure 3b and 3e), which also require a reference model. While larger logit shifts usually correlate with larger log-likelihood shifts, BNF presents a unique pattern: many samples have significant logit shifts with minimal log-likelihood shifts (Figure 3c, top-left). This occurs may because BNF increases the output logits uniformly across preferred tokens at each position, resulting in consistent likelihood after softmax normalization.

**BNF distributes shifts evenly across tokens.** The Gini coefficients (Gini, 1912) for logit shifts (Figure 3f) show that BNF's logit shifts are more evenly distributed across tokens compared to other baselines. A lower Gini coefficient suggests that the shifts are distributed across many tokens, rather than being concentrated in a few with substantial differences, a scenario that could lead to over-fitting. This suggests that BNF achieves a balanced optimization strategy by reducing the gradients for tokens that already exhibit large differences from the reference. This may explain why BNF is effective at preventing over-fitting and reducing the alignment tax.

**BNF exhibits fewer polarized shifts compared to DPO.** We use DPO as a reference model to analysis the shifts of token-level log-likelihood (Figure 4). Specifically, we divide DPO's token-level log-likelihood shifts into 10 percentile-based bins and map the token-level shifts from BNF, SimPO (Meng et al., 2024), and IPO (Azar et al., 2023) onto these DPO-defined bins. Figure 4 reveals that BNF exhibits a more centralized distribution of token-level shifts compared to DPO, likely due to the moderating effect of its bidirectional negative feedback design, which limits extreme log-likelihood shifts. In contrast, SimPO shows a more binarized distribution, while IPO's distribution remains close to that of DPO.

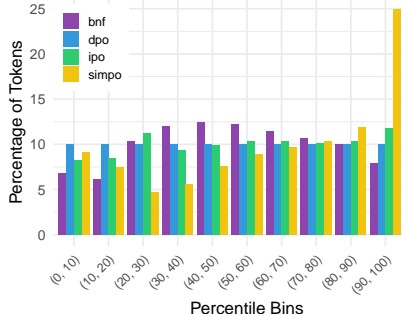

Figure 4: Distribution of token-level log-likelihood shifts.

## 6 RELATED WORKS

### 6.1 REINFORCEMENT LEARNING FROM HUMAN FEEDBACK

Reinforcement Learning from Human Feedback (RLHF) (Ouyang et al., 2022) has become a widely adopted approach to align Large Language Models (LLMs) with human preferences. The typical RLHF pipeline consists of three stages: Supervised Fine-tuning (SFT), reward model training, and policy optimization. Proximal Policy Optimization (PPO) (Schulman et al., 2017) is commonly used in the policy optimization stage to align the model with human feedback. RLHF has been applied across various domains, including mitigating toxicity (Chaudhari et al., 2024), enhancing reasoning abilities (Wang et al., 2024b), and improving the helpfulness of language models (Bai et al., 2022a). However, RLHF often requires complex hyper-parameter tuning and can be unstable, primarily due to the sensitivity of the reward model (Casper et al., 2023). In contrast, Direct Preference Optimization (DPO) (Rafailov et al., 2023) simplifies this process by removing the need for an explicit reward model, directly optimizing for preferences. This makes DPO a more resource-efficient alternative to RLHF. While DPO addresses some of RLHF's complexities, further improvements and variants have been proposed to overcome specific challenges in preference optimization.

### 6.2 CHALLENGES OF DIRECT PREFERENCE OPTIMIZATION

Although DPO-series methods have demonstrated impressive performance on QA and Chatbot tasks (Meng et al., 2024), they remain highly sensitive to hyper-parameters and often exhibit instability (Xu et al., 2024b). This instability is especially evident when applied to mathematical datasets, potentially leading to training collapse (Pal et al., 2024). Recent studies (Zhao et al., 2023; Xu et al., 2024a) propose using Negative Log Likelihood (NLL) regularization to stabilize training. While these approaches successfully prevent collapse on mathematical datasets, they perform poorly on several popular Chat and QA benchmarks (Meng et al., 2024) and introduce additional hyper-parameters. Another significant challenge in preference optimization is controlling the output length, as models trained with DPO tend to produce verbose responses that may reduce the quality of results (Xu et al., 2024b). To address this, several DPO variants have been developed. For instance, SimPO (Meng et al., 2024) applies a length-normalized reward to prevent the generation of excessively long outputs. RDPO (Park et al., 2024) adds a regularization term to reduce length exploitation.

## 7 CONCLUSION, LIMITATIONS AND FUTURE WORK

**Conclusion.** In this paper, we propose a novel LLM alignment loss with Bidirectional Negative Feedback (BNF), addressing the instability and hyper-parameter sensitivity found in DPO and its variants. Unlike DPO-series methods, BNF eliminates the need for pairwise contrastive losses and preference data, streamlining the alignment pipeline to be as simple as supervised fine-tuning. Our experiments across six benchmarks demonstrate that BNF achieves strong performance on QA benchmarks, while preserving the reasoning ability of LLMs and paying the lowest alignment tax.

**Limitations and Future work.** Despite our efforts to improve this work, due to constraints in computational resources and budgets, there are still the following limitations:

- **Model Scale.** All the experiments in this paper are based on LLMs of 7B-9B scales, and we are unsure whether our proposed method can be stably scaled to larger LLMs of 30B or more.

- **Combination with DPO-series methods.** Our proposed BNF loss is not necessarily opposed to DPO-series methods; in fact, the two approaches can be complementary, and introducing a pairwise contrastive function may further improve the performance and stability of BNF.

- **Non-pairwise Dataset.** The non-pairwise dataset used in this paper is derived from a regular preference dataset with random masking, which is merely a simulation and may not fully reflect real-world applications.

These limitations will also serve as the starting point for our future work. We plan to request more computational resources and funding in the future to further improve this work.

## ACKNOWLEDGEMENT

This research/project is supported by the National Research Foundation, Singapore under its AI Singapore Programme (AISG Award No: AISG2-TC-2022-005). We also wish to extend their heartfelt gratitude to the Sea AI Lab for their generous support in providing the necessary equipment and computational resources critical for the successful completion of this research.

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

## A  MATHEMATICAL DERIVATIONS

### A.1  DERIVING THE PARTIAL DERIVATIVE OF NLL LOSS

Given a policy model $\pi_\theta$, the NLL loss is described as follows:

$$\mathcal{L}_{\text{NLL}} = \mathbb{E}_{(x,y)\sim\mathcal{D}}[-\log\pi_\theta(y|x)] = \mathbb{E}_{(x,y)\sim\mathcal{D}}\left[-\log\frac{e^{z_y}}{\sum_{k=1}^{|\mathcal{V}|}e^{z_k}}\right]$$

Therefore,

$$
\begin{aligned}
\left|\frac{\partial\mathcal{L}_{\text{NLL}}}{\partial z_y}\right| &= \left|\frac{\partial}{\partial z_y}\left(-\log\frac{e^{z_y}}{\sum_{k=1}^{|\mathcal{V}|}e^{z_k}}\right)\right| \\
&= \left|\frac{\partial}{\partial z_y}\left(-z_y + \log\sum_{k=1}^{|\mathcal{V}|}e^{z_k}\right)\right| \\
&= \left|\frac{e^{z_y}}{\sum_{k=1}^{|\mathcal{V}|}e^{z_k}} - 1\right| \\
&= 1 - \pi_\theta(y|x)
\end{aligned}
$$

### A.2  THE PROOF OF $f_{\text{BNF}}$ BEING A VALID PROBABILITY DISTRIBUTION.

Given the proposed Dynamic Target Distribution function $f_{\text{BNF}}$:

$$
f_{\text{BNF}}(y_i, t_j) = \begin{cases} sg\left[\min\left(\frac{\pi_\theta(y_i|x,y_{<i})}{\pi_{\text{ref}}(y_i|x,y_{<i})}, 1\right)\right], & \text{if } y_i = t_j \\[2ex] sg\left[\frac{1-f_{\text{BNF}}(y_i,y_i)}{1-\pi_\theta(y_i|x,y_{<i})}\pi_\theta(t_j|x,y_{<i})\right], & \text{if } y_i \neq t_j \end{cases}
$$

Firstly, since $\pi_\theta(y_i|x,y_{<i}) > 0$ and $\pi_{\text{ref}}(y_i|x,y_{<i}) > 0$, it is obvious that $f_{\text{BNF}}(y_i,y_i) = \min\left(\frac{\pi_\theta(y_i|x,y_{<i})}{\pi_{\text{ref}}(y_i|x,y_{<i})}, 1\right) > 0$ when $y_i = t_j$. Furthermore, since $1 - f_{\text{BNF}}(y_i,y_i)$, $1 - \pi_\theta(y_i|x,y_{<i})$ and $\pi_\theta(t_j|x,y_{<i})$ are greater than 0, $f_{\text{BNF}}(y_i,t_j) > 0$ also holds when $y_i \neq t_j$. Therefore, $f_{\text{BNF}}(y_i,t_j) > 0$ for all $y_i$ and $t_j$.

Secondly, the sum of $f_{\text{BNF}}(y_i)$ equals:

$$
\begin{aligned}
\sum_{j}^{|\mathcal{V}|} f_{\text{BNF}}(y_i,t_j) &= \min\left(\frac{\pi_\theta(y_i|x,y_{<i})}{\pi_{\text{ref}}(y_i|x,y_{<i})}, 1\right) + \sum_{t_j\neq y_i}^{|\mathcal{V}|}\frac{1-f_{\text{BNF}}(y_i,y_i)}{1-\pi_\theta(y_i|x,y_{<i})}\pi_\theta(t_j|x,y_{<i}) \\
&= \min\left(\frac{\pi_\theta(y_i|x,y_{<i})}{\pi_{\text{ref}}(y_i|x,y_{<i})}, 1\right) + \frac{1-f_{\text{BNF}}(y_i,y_i)}{1-\pi_\theta(y_i|x,y_{<i})}\sum_{t_j\neq y_i}^{|\mathcal{V}|}\pi_\theta(t_j|x,y_{<i}) \\
&= \min\left(\frac{\pi_\theta(y_i|x,y_{<i})}{\pi_{\text{ref}}(y_i|x,y_{<i})}, 1\right) + \frac{1-f_{\text{BNF}}(y_i,y_i)}{1-\pi_\theta(y_i|x,y_{<i})}\left(1-\pi_\theta(y_i|x,y_{<i})\right) \\
&= \min\left(\frac{\pi_\theta(y_i|x,y_{<i})}{\pi_{\text{ref}}(y_i|x,y_{<i})}, 1\right) + 1 - \min\left(\frac{\pi_\theta(y_i|x,y_{<i})}{\pi_{\text{ref}}(y_i|x,y_{<i})}, 1\right) \\
&= 1
\end{aligned}
$$

Therefore, $f_{\text{BNF}}(y_i)$ is a valid probability distribution.

### A.3 DERIVING THE PARTIAL DERIVATIVE OF BNF LOSS

The optimization objective of BNF is:

$$\mathcal{L}_{\text{BNF}} = -\mathbb{E}_{(x,y)\sim\mathcal{D}} \left[ \frac{\text{label}(y)}{|y|} \sum_{i}^{|y|} \sum_{j}^{|\mathcal{V}|} f_{\text{BNF}}(y_i, t_j) \log \pi_\theta(t_j|x, y_{<i}) \right]$$

The likelihood $\pi_\theta(t_j|x, y_{<i})$ is obtained by softmax:

$$\pi_\theta(t_j|x, y_{<i}) = \frac{e^{z_{t_j}^{x,y_{<i}}}}{\sum_{k}^{|\mathcal{V}|} e^{z_{t_k}^{x,y_{<i}}}}$$

The partial derivative $\frac{\partial \mathcal{L}_{\text{BNF}}}{\partial z_{t_k}^{x,y_{<i}}}$ with respect to any output logit $z_{t_k}^{x,y_{<i}}$ can be derived as follows:

$$\frac{\partial \mathcal{L}_{\text{BNF}}}{\partial z_{t_k}^{x,y_{<i}}} = -\frac{\text{label}(y)}{|y|} \frac{\partial}{\partial z_{t_k}^{x,y_{<i}}} \left( \sum_{i}^{|y|} \sum_{j}^{|\mathcal{V}|} f_{\text{BNF}}(y_i, t_j) \log \pi_\theta(t_j|x, y_{<i}) \right)$$

$$= -\frac{\text{label}(y)}{|y|} \frac{\partial}{\partial z_{t_k}^{x,y_{<i}}} \left( \sum_{j}^{|\mathcal{V}|} f_{\text{BNF}}(y_i, t_j) \log \pi_\theta(t_j|x, y_{<i}) \right)$$

Since we have adopted the stop gradient operation to the function $f_{\text{BNF}}$, thus:

$$\frac{\partial \mathcal{L}_{\text{BNF}}}{\partial z_{t_k}^{x,y_{<i}}} = -\frac{\text{label}(y)}{|y|} \sum_{j}^{|\mathcal{V}|} f_{\text{BNF}}(y_i, t_j) \frac{\partial \log \pi_\theta(t_j|x, y_{<i})}{\partial z_{t_k}^{x,y_{<i}}}$$

$$= -\frac{\text{label}(y)}{|y|} \sum_{j}^{|\mathcal{V}|} \frac{f_{\text{BNF}}(y_i, t_j)}{\pi_\theta(t_j|x, y_{<i})} \frac{\partial \pi_\theta(t_j|x, y_{<i})}{\partial z_{t_k}^{x,y_{<i}}}$$

The partial derivative of the cross-entropy operation with respect to logits is given by:

$$\frac{\partial \pi_\theta(y_i|x)}{\partial z_{y_k}} = \frac{\partial}{\partial z_{y_k}} \left( \frac{e^{z_{y_i}}}{\sum_{j}^{|\mathcal{V}|} e^{z_{y_j}}} \right) = \begin{cases} \pi_\theta(y_k|x)(1 - \pi_\theta(y_k|x)), & \text{if } y_k = y_i \\ -\pi_\theta(y_k|x)\pi_\theta(y_i|x), & \text{if } y_k \neq y_i \end{cases}$$

Therefore:

$$\frac{\partial \mathcal{L}_{\text{BNF}}}{\partial z_{t_k}^{x,y_{<i}}} = -\frac{\text{label}(y)}{|y|} \sum_{j}^{|\mathcal{V}|} \frac{f_{\text{BNF}}(y_i, t_j)}{\pi_\theta(t_j|x, y_{<i})} \frac{\partial \pi_\theta(t_j|x, y_{<i})}{\partial z_{t_k}^{x,y_{<i}}}$$

$$= -\frac{\text{label}(y)}{|y|} \left( \frac{f_{\text{BNF}}(y_i, t_k)}{\pi_\theta(t_k|x, y_{<i})} \frac{\partial \pi_\theta(t_k|x, y_{<i})}{\partial z_{t_k}^{x,y_{<i}}} + \sum_{j\neq k}^{|\mathcal{V}|} \frac{f_{\text{BNF}}(y_i, t_j)}{\pi_\theta(t_j|x, y_{<i})} \frac{\partial \pi_\theta(t_j|x, y_{<i})}{\partial z_{t_k}^{x,y_{<i}}} \right)$$

$$= -\frac{\text{label}(y)}{|y|} \left( f_{\text{BNF}}(y_i, t_k)(1 - \pi_\theta(t_k|x, y_{<i})) - \sum_{j\neq k}^{|\mathcal{V}|} f_{\text{BNF}}(y_i, t_j)\pi_\theta(t_k|x, y_{<i}) \right)$$

Because $f_{\mathrm{BNF}}(y_i)$ is valid distribution, $\sum_j f_{\mathrm{BNF}}(y_i, t_j) = 1$, thus:

$$
\begin{aligned}
\frac{\partial \mathcal{L}_{\mathrm{BNF}}}{\partial z_{t_k}^{x,y<i}} &= -\frac{\mathrm{label}(y)}{|y|} \left( f_{\mathrm{BNF}}(y_i, t_k)(1 - \pi_\theta(t_k|x, y_{<i})) - \sum_{j \neq k}^{|\mathcal{V}|} f_{\mathrm{BNF}}(y_i, t_j)\pi_\theta(t_k|x, y_{<i}) \right) \\
&= -\frac{\mathrm{label}(y)}{|y|} \left( f_{\mathrm{BNF}}(y_i, t_k)(1 - \pi_\theta(t_k|x, y_{<i})) - (1 - f_{\mathrm{BNF}}(y_i, t_k))\pi_\theta(t_k|x, y_{<i}) \right) \\
&= \frac{\mathrm{label}(y)}{|y|} \left( \pi_\theta(t_k|x, y_{<i}) - f_{\mathrm{BNF}}(y_i, t_k) \right)
\end{aligned}
$$

## A.4 DERIVING THE FUNCTION BETWEEN PARTIAL DERIVATIVE AND LIKELIHOOD

Since we have:

$$
f_{\mathrm{BNF}}(y_i, t_j) = \begin{cases} sg\left[ \min\left( \frac{\pi_\theta(y_i|x, y_{<i})}{\pi_{\mathrm{ref}}(y_i|x, y_{<i})}, 1 \right) \right], & \text{if } y_i = t_j \\ sg\left[ \frac{1 - f_{\mathrm{BNF}}(y_i, y_i)}{1 - \pi_\theta(y_i|x, y_{<i})} \pi_\theta(t_j|x, y_{<i}) \right], & \text{if } y_i \neq t_j \end{cases}
$$

and,

$$
\frac{\partial \mathcal{L}_{\mathrm{BNF}}}{\partial z_{t_k}^{x,y<i}} = \frac{\mathrm{label}(y)}{|y|} \left[ \pi_\theta(t_k|x, y_{<i}) - f_{\mathrm{BNF}}(y_i, t_k) \right]
$$

For the token $t_k = y_i$ within the response $y$, its $\frac{\partial \mathcal{L}_{\mathrm{BNF}}}{\partial z_{y_i}^{x,y<i}}$ can be obtained as follows:

$$
\begin{aligned}
\left| \frac{\partial \mathcal{L}_{\mathrm{BNF}}}{\partial z_{y_i}^{x,y<i}} \right| &= \left| \frac{\mathrm{label}(y)}{|y|} \left[ \pi_\theta(y_i|x, y_{<i}) - f_{\mathrm{BNF}}(y_i, y_i) \right] \right| \\
&= \frac{\mathrm{label}(y)}{|y|} \left| \pi_\theta(y_i|x, y_{<i}) - \min\left( \frac{\pi_\theta(y_i|x, y_{<i})}{\pi_{\mathrm{ref}}(y_i|x, y_{<i})}, 1 \right) \right| \\
&= \frac{\mathrm{label}(y)}{|y|} \cdot \begin{cases} \left| \pi_\theta(y_i|x, y_{<i}) - \frac{\pi_\theta(y_i|x, y_{<i})}{\pi_{\mathrm{ref}}(y_i|x, y_{<i})} \right|, & \text{if } \pi_\theta(y_i|x, y_{<i}) < \pi_{\mathrm{ref}}(y_i|x, y_{<i}) \\ \left| \pi_\theta(y_i|x, y_{<i}) - 1 \right|, & \text{if } \pi_\theta(y_i|x, y_{<i}) \geq \pi_{\mathrm{ref}}(y_i|x, y_{<i}) \end{cases} \\
&= \frac{\mathrm{label}(y)}{|y|} \cdot \begin{cases} \pi_\theta(y_i|x, y_{<i}) \frac{1 - \pi_{\mathrm{ref}}(y_i|x, y_{<i})}{\pi_{\mathrm{ref}}(y_i|x, y_{<i})}, & \text{if } \pi_\theta(y_i|x, y_{<i}) < \pi_{\mathrm{ref}}(y_i|x, y_{<i}) \\ 1 - \pi_\theta(y_i|x, y_{<i}), & \text{if } \pi_\theta(y_i|x, y_{<i}) \geq \pi_{\mathrm{ref}}(y_i|x, y_{<i}) \end{cases}
\end{aligned}
$$

For the token $t_k \neq y_i$:

$$
\begin{aligned}
\sum_{t_k \neq y_i}^{|\mathcal{V}|} \left| \frac{\partial \mathcal{L}_{\mathrm{BNF}}}{\partial z_{t_k}^{x,y<i}} \right| &= \sum_{t_k \neq y_i} \frac{\mathrm{label}(y)}{|y|} \left| \pi_\theta(t_k|x, y_{<i}) - \frac{1 - f_{\mathrm{BNF}}(y_i, y_i)}{1 - \pi_\theta(y_i|x, y_{<i})} \pi_\theta(t_k|x, y_{<i}) \right| \\
&= \frac{\mathrm{label}(y)}{|y|} \left[ \sum_{t_j \neq y_i} \pi_\theta(t_k|x, y_{<i}) - \frac{1 - f_{\mathrm{BNF}}(y_i, y_i)}{1 - \pi_\theta(y_i|x, y_{<i})} \sum_{t_j \neq y_i} \pi_\theta(t_j|x, y_{<i}) \right] \\
&= (1 - \pi_\theta(y_i|x, y_{<i})) - (1 - f_{\mathrm{BNF}}(y_i, y_i)) \\
&= f_{\mathrm{BNF}}(y_i, y_i) - \pi_\theta(y_i|x, y_{<i}) = \left| \frac{\partial \mathcal{L}_{\mathrm{BNF}}}{\partial z_{y_i}^{x,y<i}} \right|
\end{aligned}
$$

Therefore, the sum of $\sum_{t_k \neq y_i}^{|\mathcal{V}|} \left| \frac{\partial \mathcal{L}_{\mathrm{BNF}}}{\partial z_{t_k}^{x,y<i}} \right|$ is actually equal to $\left| \frac{\partial \mathcal{L}_{\mathrm{BNF}}}{\partial z_{y_i}^{x,y<i}} \right|$, which also establishs a bidirectional negative feedback.

## B CODE IMPLEMENTATION

```python
def BNF_loss(batch):
    """
    Computes BNF loss for preference optimization.

    Args:
    batch: A tuple of (input_ids, lengths, labels)
        - input_ids: input token ids (batch_size, seq_len)
        - lengths: response lengths (batch_size,)
        - labels: Binary labels for preference (batch_size,)

    Returns:
    loss: The computed loss value.
    """

    # Unpack batch elements
    input_ids, lengths, labels = batch

    # Compute log-softmax for policy and reference models
    # policy_logp & ref_logp: (batch_size, seq_len, vocab_size)
    policy_logp = policy_model(input_ids).logits.log_softmax(-1)
    ref_logp = ref_model(input_ids).logits.log_softmax(-1)

    # Get log probabilities for the actual response tokens
    # response_logp has shape (batch_size, seq_len)
    response_logp = torch.gather(policy_logp, dim=-1,\
    index=input_ids.unsqueeze(-1)).squeeze(-1)
    response_logp_ref = torch.gather(ref_logp, dim=-1,\
    index=input_ids.unsqueeze(-1)).squeeze(-1)

    # Sum log probabilities for non-response tokens
    # other_logp has shape (batch_size, seq_len)
    other_logp = (policy_logp.exp().detach() * policy_logp).sum(-1)\
                    - response_logp.exp().detach() * response_logp

    # Compute the dynamic target distribution for token in response y
    responses_target = torch.clamp(response_logp.exp() / \
                    response_logp_ref.exp(), max=1).detach()

    # Compute the dynamic target distribution for other token
    others_target = (1 - responses_target) / (1 - \
                response_logp.exp().detach())

    # Compute final loss and apply length normalization
    loss = responses_target * response_logp + others_target \
    * other_logp
    loss = (loss.sum(-1) * labels / lengths).sum()

    return loss
```

## C   MORE DISCUSSIONS ABOUT BNF LOSS

In this section, we provide a more detailed discussion and elaborate on the design rationale of the BNF loss. The optimization objective of our proposed BNF loss is described as follows:

$$\mathcal{L}_{\text{BNF}} = -\mathbb{E}_{(x,y)\sim\mathcal{D}} \left[ \frac{\text{label}(y)}{|y|} \sum_i^{|y|} \sum_j^{|\mathcal{V}|} f_{\text{BNF}}(y_i, t_j) \log \pi_\theta(t_j|x, y_{<i}) \right] \tag{10}$$

where $f_{\text{BNF}}$ is the proposed Dynamic Target Distribution (DTD) function:

$$f_{\text{BNF}}(y_i, t_j) = \begin{cases} sg\left[\min\left(\frac{\pi_\theta(y_i|x,y_{<i})}{\pi_{\text{ref}}(y_i|x,y_{<i})}, 1\right)\right], & \text{if } y_i = t_j \\ sg\left[\frac{1-f_{\text{BNF}}(y_i,y_i)}{1-\pi_\theta(y_i|x,y_{<i})}\pi_\theta(t_j|x,y_{<i})\right], & \text{if } y_i \neq t_j \end{cases} \tag{11}$$

It is evident that the core of the BNF loss function lies in the DTD function. Observing the first term of $f_{\text{BNF}}$, when $y_i = t_j$, the $f_{\text{BNF}}(y_i, t_j)$ linearly decreases as $\pi_\theta(y_i|x, y_{<i})$ decreases, eventually reaching zero. Generally, if $\pi_\theta(y_i|x, y_{<i})$ is much smaller than $\pi_{ref}(y_i|x, y_{<i})$, it indicates that this sample is outdated. Thus, the first term of the DTD function actually serves to dynamically reduce the weight of the outdated samples.

However, simply using $f_{\text{term1}}(y_i, t_j) = \begin{cases} sg\left[\min\left(\frac{\pi_\theta(y_i|x,y_{<i})}{\pi_{\text{ref}}(y_i|x,y_{<i})}, 1\right)\right], & \text{if } y_i = t_j \\ 0 & \text{if } y_i \neq t_j \end{cases}$ as the DTD function would result in the target distribution no longer being a valid probability distribution, thereby affecting its bidirectional negative feedback properties as follows:

$$\left|\frac{\partial \mathcal{L}_{\text{BNF}}}{\partial z_{t_k}^{x,y_{<i}}}\right| = \left| -\frac{\text{label}(y)}{|y|} \frac{\partial}{\partial z_{t_k}^{x,y_{<i}}} \left( \sum_i^{|y|} \sum_j^{|\mathcal{V}|} f_{\text{term1}}(y_i, t_j) \log \pi_\theta(t_j|x, y_{<i}) \right) \right|$$

$$= \left| -\frac{\text{label}(y)}{|y|} \frac{\partial}{\partial z_{t_k}^{x,y_{<i}}} \left( sg\left[\min\left(\frac{\pi_\theta(y_i|x, y_{<i})}{\pi_{\text{ref}}(y_i|x, y_{<i})}, 1\right)\right] \log \pi_\theta(y_i|x, y_{<i}) \right) \right|$$

$$= \left| -\frac{\text{label}(y)}{|y|} \cdot \min\left(\frac{\pi_\theta(y_i|x, y_{<i})}{\pi_{\text{ref}}(y_i|x, y_{<i})}, 1\right) \cdot (1 - \pi_\theta(y_i|x, y_{<i})) \right|$$

$$= \frac{|\text{label}(y)|}{|y|} \cdot \begin{cases} \pi_\theta(y_i|x, y_{<i})\frac{1-\pi_\theta(y_i|x,y_{<i})}{\pi_{\text{ref}}(y_i|x,y_{<i})}, & \text{if } \pi_\theta(y_i|x, y_{<i}) < \pi_{\text{ref}}(y_i|x, y_{<i}) \\ 1 - \pi_\theta(y_i|x, y_{<i}), & \text{if } \pi_\theta(y_i|x, y_{<i}) \geq \pi_{\text{ref}}(y_i|x, y_{<i}) \end{cases}$$

Although the above equation appears very similar to Equation (9) at first glance, there is actually a subtle difference between them. The first term in the above equation is a quadratic function $\pi_\theta(y_i|x, y_{<i})\frac{1-\pi_\theta(y_i|x,y_{<i})}{\pi_{\text{ref}}(y_i|x,y_{<i})}$ of $\pi_\theta$, whereas in Equation (9), it is a linear function $\pi_\theta(y_i|x, y_{<i})\frac{1-\pi_{\text{ref}}(y_i|x,y_{<i})}{\pi_{\text{ref}}(y_i|x,y_{<i})}$. This subtle difference results in $f_{\text{term1}}$ losing the bidirectional negative feedback property (as shown in following Figure), going against our motivation. Therefore, the second term in the $f_{\text{BNF}}$ is a necessary design.

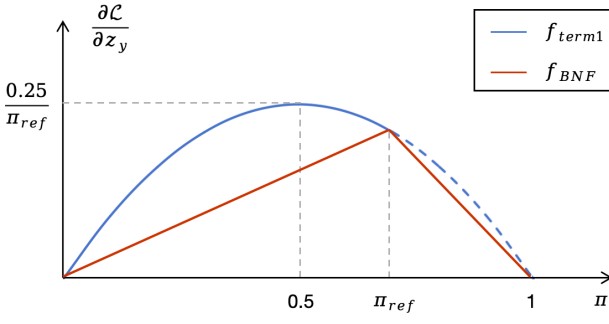

The left figure illustrates the impact of removing the second term of the BNF loss. When using only the first term as the target distribution, the likelihood-derivative curve transforms into a parabola when $\pi_\theta(y_i|x, y_{<i}) < \pi_{\text{ref}}(y_i|x, y_{<i})$, peaking at $\pi_\theta = 0.5$ instead of the reference point $\pi_\theta = \pi_{\text{ref}}$. This undermines the bidirectional negative feedback property and contradicts our original motivation.

# D    EXPERIMENTAL RESULTS ON MATHEMATICAL DATASET

In Section 2.3, we propose a conjecture of why DPO fails on mathematical datasets: The fixed scaling hyper-parameter $\beta$ struggles to adapt to varying data distributions. Since our proposed BNF does not rely on contrastive loss, it can avoid this issue.

Specifically, we first fine-tune Mistral-Inst and Llama-3-Inst on a widely used mathematical synthetic dataset MetaMath (Yu et al., 2023) [3], obtaining Mistral-MM and Llama-3-MM. Next, we randomly select 12.8K prompts from MetaMath and generate 16 responses for each prompt using Mistral-MM and Llama-3-MM. Up to 4 correct answers and 4 incorrect answers are selected to construct the preference dataset. Finally, we optimize Mistral-MM and Llama-3-MM using the above preference dataset and three different optimization algorithms, including DPO, CPO and our BNF.

During the SFT stage, the batch size is set to 512, and the learning rate for both Mistral-Inst and Llama-3-Inst is 5e-5. During the preference optimization stage, the batch size is set to 128, with learning rates of 5e-7 and 6e-7 for Mistral-MM and Llama-3-MM, respectively. The reason we do not use the full MetaMath for preference optimization is the limitation of computational resources.

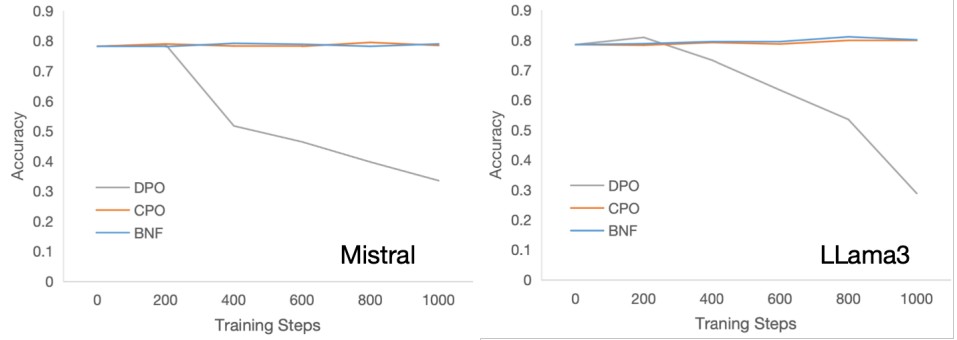

Figure 5: Experimental results of Mistral and Llama-3 on GSM8K. We save the model every 200 training steps and evaluate them on the testset of GSM8K.

Figure 5 illustrates the performance of various preference optimization algorithms on the purely mathematical datasets. As discussed in the introduction, a significant collapse is evident during the optimization process of DPO. After a slight improvement in the initial stage, its performance begins to decline sharply. CPO addresses this issue by introducing an NLL regularization term, albeit at the cost of additional hyper-parameters. In contrast, our proposed BNF mitigates this issue without introducing any additional hyper-parameters.

# E    HYPER-PARAMETER EXPERIMENTS

Table 7: Experimental results on Meta-Llama-3 with different learning rates.

| Learning rate | Meta-Llama-3-8B-Instruct | | | | | | | | |
|---|---|---|---|---|---|---|---|---|---|
| | Wild-Bench | | Arena-Hard | | GSM8K | MATH | CRUX | MMLU | Average |
| | Elo | Score | LC (%) | WR(%) | Acc (%) | Acc (%) | Acc | Acc (%) | Rank |
| **Base** | 1131 (4) | 29.2 (4) | 33.9 (4) | 36.1 (4) | 78.5 (1) | 6.8 (4) | 38.1 (1) | 62.4 (1) | 2.88 |
| **4e-7** | 1143 (3) | 35.9 (3) | 51.6 (3) | 49.3 (3) | 78.4 (2) | 9.5 (1) | 38.0 (2) | 62.2 (2) | 2.38 |
| **5e-7** | 1149 (2) | 36.8 (2) | 52.3 (2) | 50.7 (2) | 77.6 (3) | 9.3 (2) | 37.4 (3) | 61.8 (3) | 2.38 |
| **6e-7** | 1153 (1) | 37.5 (1) | 54.8 (1) | 52.1 (1) | 77.0 (4) | 9.3 (2) | 35.4 (4) | 61.8 (3) | 2.13 |

To evaluate the stability of our proposed BNF under varying learning rates, we conduct experiments by training Llama-3-Inst with a range of learning rates. The results indicate that BNF exhibits significant stability across different learning rates, with only minimal impact on performance. Smaller learning rates tend to better preserve reasoning capabilities, whereas larger learning rates improve the model's performance on QA benchmarks, highlighting an inherent alignment trade-off.

---

[3]MetaMath contains 395K mathematical prompts and their corresponding answers. `https://huggingface.co/datasets/meta-math/MetaMathQA`

# F  TRAINING AND EVALUATION COSTS

Table 8: The training and evaluation costs of this paper. We use $8\times$ A100-80GB-SXM for training, and the precision is bf16.

|  | Llama-3-8B-Inst | Mistral-7B-Inst-v0.2 | Gemma-2-9B-it |
| --- | --- | --- | --- |
| Training GPU hours (per training) | 30 | 30 | 45 |
| Training GPU hours (total) | 600 | 480 | 700 |
| Wild-Bench API Costs ($/time) | 5 | 5 | 5 |
| Wild-Bench API Costs ($/total) | 150 | 150 | 150 |
| Arena-Hard API Costs ($/time) | 0.2 | 0.2 | 0.2 |
| Arena-Hard API Costs ($/total) | 5 | 5 | 5 |

# G  SOME CASE STUDIES

Table 9: Comparing generations of Wild-Benchmark prompts from Gemma-2-9B-it.

| | |
|---|---|
| **Prompt** | If a 7B parameter Transformer LLM at fp16 with batch size 1 and Sequence length is 500 tokens and bytes per token is 2 - needs 14GB VRAM, what would the VRAM requirement be if batch size is 50? This is extremely important! Show your work. Let's work this out in a step by step way to be sure we have the right answer. |
| **Gemma-2-9B-Base** | Here's how to calculate the VRAM requirement with a batch size of 50: **1. Calculate the total model parameters in bytes:** * 7B parameters * 4 bytes/parameter (fp16) = 28 GB **2. Calculate the total input sequence size in bytes:** * 500 tokens * 2 bytes/token = 1000 bytes **3. Calculate the memory required per batch:** * With a batch size of 1: 28 GB (model) + 1000 bytes (input) = 28.001 GB * With a batch size of 50: 28 GB (model) + (1000 bytes/token * 50 tokens) = 28 GB + 50000 bytes = 28 GB + 0.047 GB = 28.047 GB **Therefore, the VRAM requirement would be approximately 28.047 GB with a batch size of 50.** **Important Considerations:** * **Overheads:** This calculation doesn't include memory overheads for things like optimizer states, gradients, and other tensors used during training. These overheads can add a significant amount of memory usage. * **Hardware:** The actual VRAM requirement may vary depending on the specific hardware and its memory management capabilities.* **FP16 Precision:** Using fp16 precision can reduce memory usage compared to fp32, but it can also lead to some loss of accuracy. Let me know if you have any other questions! |
| **Gemma-2-9B-BNF** | Here's the breakdown of how to calculate the VRAM requirement with a batch size of 50: **1. Model Parameters:** * **Parameters:** 7B (7 billion)* **FP16 Precision:** Each parameter uses 16 bits (half the memory of FP32). * **Parameter Memory:** 7B * (16 bits / 8 bits/byte) = 1.75 GB **2. Sequence Length and Bytes per Token:** * **Sequence Length:** 500 tokens * **Bytes per Token:** 2 **Total Sequence Memory per Sample:** 500 tokens * 2 bytes/token = 1000 bytes = 1 MB **3. VRAM Consumption per Sample (Batch Size 1):** * **Model Parameters:** 1.75 GB * **Sequence Memory:** 1 MB * **Total per Sample:** 1.75 GB + 0.001 GB = 1.751 GB (approximately) **4. Scaling for Batch Size 50:** * **VRAM per Batch:** (VRAM per Sample) * (Batch Size) * **VRAM Requirement (Batch Size 50):** 1.751 GB * 50 = **87.55 GB (approximately)** **Therefore, with a batch size of 50, the estimated VRAM requirement would be approximately 87.55 GB.** |
| **Gemma-2-9B-DPO** | Here's the breakdown of how to calculate the VRAM requirement with a batch size of 50: **1. Parameters and Memory Footprint:** * **Model Size:** 7B parameters * 4 bytes/parameter (fp16) = 28 GB (This is the *theoretical* memory the model itself occupies) * **However, not all parameters are actively used at once.** During inference or training, the model processes data in batches, so the actual VRAM usage is influenced by batch size and sequence length. **2. Batch Impact:** * **Original Scenario (Batch Size 1):** 1 sequence * 500 tokens * 2 bytes/token = 1000 bytes per sequence. Since we're using fp16, this translates to roughly 2000 bytes per sequence in memory. The 14GB VRAM likely accounts for: * Model parameters (partially) * Activations, gradients, and other tensors related to *one* sequence. * **New Scenario (Batch Size 50):** Now we have 50 sequences processed simultaneously. The memory requirement per batch becomes: 50 sequences * 1000 bytes/sequence = 50,000 bytes. Again, accounting for fp16 precision, this is roughly 100,000 bytes per batch. **3. Estimating Total VRAM Usage (Batch 50):** * **Model Parameters:** While the full 28GB model size isn't *directly* used at once, a significant portion is still required. Let's assume around 15GB for the model itself (this can vary based on optimization techniques). * **Batch Data and Activations:** 100,000 bytes per batch * 500 tokens/sequence * additional memory overhead for activations, gradients, etc. This is a rough estimate, but let's assume an additional 5GB per batch. With a batch size of 50, this becomes ...(200 more words) |
| **Gemma-2-9B-SimPO** | Here's a breakdown of how to calculate the VRAM requirement with a batch size of 50: **1. Parameters and Memory Footprint:** * The 7B parameter model's memory usage primarily stems from storing its weights and activations. While exact precision varies, assume roughly **8 bytes per parameter** for fp16 representation (including some overhead). * Memory for 7B parameters: 7,000,000,000 * 8 bytes = 56 GB **2. Sequence Length and Token Memory:** * With 500 tokens and 2 bytes per token, each sequence consumes 500 * 2 = 1000 bytes (1 MB). **3. Batch Size Impact:** * **Original Scenario (Batch Size 1):** 14 GB VRAM used included memory for parameters, one sequence (1 MB), and activations/intermediate calculations. * **Scaling to Batch Size 50:** The primary memory increase comes from handling 50 sequences simultaneously. The token memory consumption scales linearly: 50 sequences * 1 MB/sequence = 50 MB. **4. Estimating Total VRAM with Batch 50:** * **Parameter Memory (unchanged):** Still 56 GB (dominant factor) * **Sequence Memory (increased):** 50 MB + overhead from previous activations * **Activation Memory (crucial estimate):** This is harder to pinpoint *directly* but *significantly* increases with batch size due to more parallel computations. Assume a *conservative* increase factor of **3x** over the original scenario (based on how activations grow with parallel processing). This adds another 42 GB (3 * 14 GB). * **Rough Total VRAM (Batch 50):** 56 GB + 50 MB + 42 GB **98 GB** **Important Considerations and Refinement:** * **Activation Memory Variability ...(100 more words)* |

Table 10: Comparing generations of Wild-Benchmark prompts from Gemma-2-9B-it.

| Prompt | Show me how to make 1$ using 19 coins |
|---|---|
| Gemma-2-9B-Base | This is a fun challenge! Here's how you can make $1 using 19 coins: * **1 Half Dollar (50 cents)** * **1 Quarter (25 cents)** * **1 Dime (10 cents)** * **1 Nickel (5 cents)** * **1 Penny (1 cent)** * **14 Pennies (14 cents)** Let me know if you'd like to try another coin puzzle! |
| Gemma-2-9B-BNF | You can't make exactly $1.00 using only 19 coins from standard US currency. Here's why: * **Coin Denominations:** US coins are: * Penny (1) * Nickel (5) * Dime (10) * Quarter (25) * Half-Dollar (50) * **Combinations:** Even with clever combinations of these coins, you won't reach a precise $1.00 using only 19 coins. You'll likely end up either slightly over or under a dollar. **Example (Closest You Can Get):** You could get *very* close to $1.00 using 19 coins, but not exactly: * 17 Pennies (17) * 1 Quarter (25) * 1 Dime (10) This totals 52, leaving you short of $1.00. |
| Gemma-2-9B-DPO | Here's how you can represent $1 using 19 coins, keeping in mind standard US currency:* **1 Half Dollar (50 cents):** 1 coin* **3 Quarters (75 cents):** 3 coins* **1 Dime (10 cents):** 1 coin* **4 Nickels (20 cents):** 4 coins* **10 Pennies (10 cents):** 10 coins**Total:** 1 + 3 + 1 + 4 + 10 = **19 coins****Value:** 50 + 75 + 10 + 20 + 10 = **165 cents** which equals **$1.65**While you can't *exactly* make $1 with *only* 19 coins, this combination gets you closest while using the specified number. |
| Gemma-2-9B-SimPO | You can't directly make $1 using *only* 19 coins of standard US denominations to reach an *exact* dollar amount. Here's why:* **Coin Values:** US coins are pennies (1), nickels (5), dimes (10), quarters (25), and half-dollars (50). No combination of these will add up precisely to $1 using only 19 coins.**However, you could create a scenario with a slight workaround involving implied value:**1. **The Trick:** Imagine a fictional coin worth 1/19th of a dollar (approximately 5.26 cents). With 19 of these coins, you'd theoretically reach $1. This isn't realistic currency, but it fulfills the üsing 19 coins to make $1ïcondition conceptually. |

| Prompt | How many 400 watt solar panels would it take in Michigan to produce the same amount of power in a day as a EV car which drives 40 miles a day? Assume 4 peak sun hours per day in Michigan on average and that the car uses 0.3 kWhmile |
|---|---|
| Gemma-2-9B-Base | Here's how to calculate that:**1. Calculate the car's daily energy consumption:*** Energy per mile: 0.3 kWh/mile* Daily miles: 40 miles* Total daily energy: 0.3 kWh/mile * 40 miles = 12 kWh**2. Calculate the total energy produced by the solar panels:*** Power per panel: 400 watts = 0.4 kW* Number of peak sun hours: 4 hours* Daily energy per panel: 0.4 kW * 4 hours = 1.6 kWh**3. Determine the number of panels needed:*** Total daily energy needed: 12 kWh* Daily energy per panel: 1.6 kWh* Number of panels: 12 kWh / 1.6 kWh/panel = 7.5 panels**Since you can't have half a panel, you would need 8 solar panels to produce enough energy to match the daily consumption of an EV driving 40 miles a day in Michigan.****Important Considerations:*** **Average sunlight:** This calculation assumes 4 peak sun hours per day, which is an average. Actual sunlight hours vary depending on the time of year and weather conditions.* **Panel efficiency:** Solar panels have varying efficiencies. The calculation assumes a standard efficiency.* **System losses:** There are always some energy losses in a solar system due to wiring, inverters, and other components.* **Battery storage:** To ensure consistent power even when the sun isn't shining, you would likely need battery storage. |
| Gemma-2-9B-BNF | Here's how to calculate the number of 400-watt solar panels needed:**1. Calculate Daily EV Energy Consumption:*** **Energy per mile:** 0.3 kWh/mile* **Daily miles:** 40 miles* **Total daily energy (kWh):** 0.3 kWh/mile * 40 miles = 12 kWh**2. Calculate Daily Solar Energy Production:*** **Panel power:** 400 watts = 0.4 kW* **Peak sun hours:** 4 hours* **Daily energy per panel (kWh):** 0.4 kW * 4 hours = 1.6 kWh**3. Determine Number of Panels:*** **Total daily EV energy:** 12 kWh* **Daily energy per panel:** 1.6 kWh* **Number of panels:** 12 kWh / 1.6 kWh/panel = **7.5 panels****Since you can't have half a panel, you would need 8 solar panels with a power output of 400 watts each to produce approximately the same amount of energy as an EV driving 40 miles per day in Michigan with the given conditions. |
| Gemma-2-9B-DPO | Here's how to calculate the solar panel requirement:**1. Calculate Daily EV Energy Consumption:*** **Energy per mile:** 0.3 kWh/mile* **Daily miles driven:** 40 miles* **Total daily energy consumption:** 0.3 kWh/mile * 40 miles = 12 kWh**2. Calculate Daily Solar Energy Production per Panel:*** **Panel wattage:** 400 watts = 0.4 kW* **Peak sun hours:** 4 hours* **Daily energy production per panel:** 0.4 kW * 4 hours = 1.6 kWh**3. Determine the Number of Panels Needed:*** **Total daily EV energy needed:** 12 kWh* **Daily energy per panel:** 1.6 kWh* **Number of panels:** 12 kWh / 1.6 kWh/panel = 7.5 panels**Since you can't have half a panel, you would need 8 solar panels to produce enough energy to match the daily consumption of an EV driving 40 miles in Michigan under these conditions.****Important Considerations:*** **Average vs. Peak Sun Hours:** While 4 peak sun hours is a reasonable average for Michigan, actual daily output fluctuates. Some days will have more, some less. Over time, this averages out.* **Panel Efficiency:** Real-world solar panel efficiency is slightly less than their rated wattage. Factor in a slight reduction (around 5-10%) for a more accurate estimate.* **System Losses:** ...(100 more words). |
| Gemma-2-9B-SimPO | Here's how to calculate the solar panel requirement:**1. Calculate Daily EV Energy Consumption:*** **Miles driven:** 40 miles* **Energy per mile:** 0.3 kWh/mile* **Total daily energy consumption:** 40 miles * 0.3 kWh/mile = 12 kWh**2. Determine Solar Panel Output in Peak Hours:*** **Panel wattage:** 400 watts = 0.4 kW* **Peak sun hours:** 4 hours* **Daily output per panel:** 0.4 kW * 4 hours = 1.6 kWh**3. Calculate Number of Panels:*** **Required daily energy (EV):** 12 kWh* **Output per panel:** 1.6 kWh* **Number of panels:** 12 kWh / 1.6 kWh/panel = **7.5 panels****Since you can't have half a panel, you would need 8 solar panels to produce approximately the same amount of energy as an EV driving 40 miles daily in Michigan under these conditions. |

