# OpenReview forum: "As Simple as Fine-tuning: LLM Alignment via Bidirectional Negative Feedback Loss"
_ICLR.cc/2025/Conference — ICLR 2025 Poster_

### Official Review · Reviewer_dQGY · 2024-11-01

**Soundness:** 3
**Presentation:** 3
**Contribution:** 3
**Rating:** 6
**Confidence:** 3

**Summary:**

This paper proposes Bidirectional Negative Feedback (BNF), an LLM alignment loss that does not rely on pairwise contrastive losses.  Consequently, it does not require pairwise data and has fewer hyper-parameters compared to DPO. The authors empirically show that the models` reasoning ability is less affected when using BNF for preference optimization. They compare BNF to previous approaches such as DPO, IPO, KTO, SLiC-HF, ORPO, SimPO on QA and reasoning benchmarks.

**Strengths:**

- BNF has fewer hyper-parameters compared to DPO. If the method is robust to hyper-parameter tuning, this could make LLM alignment more compute efficient.
- BNF does not require pairwise data.
- The baselines are diverse and strong. The hyper-parameters of the other approaches were fine-tuned (although not directly by the authors).

**Weaknesses:**

- The performance across benchmarks is still relatively close to DPO.
- Gemma lacks proper baselines as the scores were only copied from another paper.

**Questions:**

- Is BNF more sample efficient because it does not require pair-wise dataset? Is there a relationship between sample efficiency and loss/dataset type?
- Do you expect the other baselines to close in on BNF with further hyper-parameter tuning?
- How sensitive is BNF to hyper-parameter tuning? Is it possible that the hyper-parameters are harder to tune despite the fact that they are fewer?
- Is the performance difference on the benchmarks significant?

---

> ### Author Response · Authors · 2024-11-21
> **To Reviewer dQGY**
>
> Dear Reviewer,
>
> Thank you for your review! We have endeavored to address all your questions and concerns below. Please let us know if there are any aspects that we need to sufficiently clarify. If you feel that your concerns have been satisfactorily addressed, we would be grateful if you would consider revising your score. Please do not hesitate to reach out with any further questions. We value your feedback and welcome any additional queries.
>
> **Q1: Is BNF more sample efficient because it does not require pair-wise dataset? Is there a relationship between sample efficiency and loss/dataset type?**
>
> In practical LLM applications, users tend to express their opinions through binary actions like upvoting or downvoting, rather than selecting the better response from two options. Therefore, we believe that avoiding reliance on paired preference data can indeed significantly enhance the efficiency of BNF in utilizing user feedback data.
>
>
>
> **Q2: Do you expect the other baselines to close in on BNF with further hyper-parameter tuning?**
>
>  Given that SimPO has already conducted a highly rigorous and extensive grid search, we believe that the performance improvement achievable through hyperparameter tuning will be limited. Even if hyperparameter adjustments result in performance gains on a specific dataset, such gains may come at the cost of performance on other datasets, commonly referred to as alignment tax.
>
>
> **Q3:  How sensitive is BNF to hyper-parameter tuning? Is it possible that the hyper-parameters are harder to tune despite the fact that they are fewer?**
>
> Thank you for your valuable feedback. In the latest version, we presents the performance of BNF under different learning rates in **Appendix E**. Overall, BNF is not harder to tune and this loss function demonstrates stable behavior.
>
>
> **Q4: Is the performance difference on the benchmarks significant?**
>
> If we focus solely on one of the QA benchmarks or reasoning benchmarks, BNF does not significantly outperform other methods. However, from a comprehensive perspective, BNF achieves the best balance across various types of benchmarks. Moreover, the primary goal of this paper is to further simplify the training pipeline and reduce the alignment tax, rather than solely improving performance.

---

> ### Author Response · Authors · 2024-11-21
> **New Modifications !**
>
> Thanks to all the reviewers for providing numerous valuable suggestions. After carefully checking your feedback, we have made the following revisions:
>
> 1. Provide more discussion and explanation of the design rationale behind the BNF loss in **Appendix C**.
>
> 2. Evaluate the performance of different preference optimization methods on purely mathematical datasets in **Appendix D**.
>
> 3. Present the performance of BNF under different learning rates in **Appendix E**.
>
> 4. Adjust certain phrasing to make the paper more rigorous.
>
> 5. Add a preference learning baseline, CPO.
>
> Please check the new version of this paper, Thanks!

---

> > ### Comment · Reviewer_dQGY · 2024-11-24
> > **Answer to rebuttal**
> >
> > Thank you for your answers. The authors have answered all the questions emphasizing that the main benefit of BNF is its reduction of the alignment tax. I am not completely convinced by the answer to the first question. Is it possible that BNF requires fewer (non pair-wise) examples compared to the other methods that relies on pair-wise data or does it require the same amount of data?

---

> > > ### Author Response · Authors · 2024-11-24
> > > **To Reviewer dQGY**
> > >
> > > Thank you for your response! It seems we may have misunderstood your first question.
> > >
> > > In our experiments, we did not find that BNF requires less data than other methods. When using the same training datasets, BNF performs comparably to other SOTA methods like DPO and SimPo on QA datasets. As you mentioned, the advantages of BNF are no tunable hyper-parameter and less alignment tax. Therefore, we have not claimed that BNF requires less data than other methods.
> > >
> > > Hope this can resolve your concern, thanks again.

---

> > > > ### Comment · Reviewer_dQGY · 2024-11-25
> > > > **Answer**
> > > >
> > > > Thank for your answer. I do not have any further questions.

---

### Official Review · Reviewer_XXYh · 2024-11-03

**Soundness:** 3
**Presentation:** 3
**Contribution:** 3
**Rating:** 6
**Confidence:** 3

**Summary:**

This paper proposes a new LLM alignment loss to address the issues of DPO variants being sensitive to hyper-parameters and unstable in mathematical tasks. The loss can maintain Bidirectional Negative Feedback (BNF) during optimization and eliminate the need of additional hyperparameters and pairwise preference data. Finally, they evaluate the method on multiple tasks and backbones to verify its effectiveness.

**Strengths:**

1.	This paper clearly outlines the previous methods and theoretically identifies their weaknesses. The motivations of this paper are clear.
2.	In terms of these issues, the authors propose their method and validate the effectiveness through theory and extensive experiments on multiple tasks and LLMs.
3.	The structure of this paper is clear, and the equations and charts are well-presented.

**Weaknesses:**

1.	Figure 2 illustrates the challenges of creating a substantial log-likelihood gap in mathematical tasks. Could you demonstrate the advantages of your method with similar examples in mathematical tasks?
2.	I am somewhat puzzled by Eq.(6). What are the motivations behind designing the equations where $y_i \neq t_i$? It appears to be a careful design.

**Questions:**

See above weaknesses.

---

> ### Author Response · Authors · 2024-11-21
> **To Reviewer XXYh**
>
> Dear Reviewer,
>
> Thank you for your review! We have endeavored to address all your questions and concerns below. Please let us know if there are any aspects that we need to sufficiently clarify. If you feel that your concerns have been satisfactorily addressed, we would be grateful if you would consider revising your score. Please do not hesitate to reach out with any further questions. We value your feedback and welcome any additional queries.
>
> **Q1:  Figure 2 illustrates the challenges of creating a substantial log-likelihood gap in mathematical tasks. Could you demonstrate the advantages of your method with similar examples in mathematical tasks?**
>
> Your feedback is highly valuable! We have added an section of math-related experiments in **Appendix D**, exploring the performance of different alignment methods on a purely mathematical preference dataset to further validate the effectiveness of our proposed approach.
>
> **Q2:  I am somewhat puzzled by Eq.(6). What are the motivations behind designing the equations where $y_i\neq t_i$? It appears to be a careful design.**
>
> Your intuition is correct; the second term in Equation (6) is indeed deliberately designed. We designed the loss function this way because simply using the first term would result in the target distribution no longer being a valid probability distribution, thereby affecting its bidirectional negative feedback properties. We have added a section in **Appendix C** to explain this in detail in the latest version of our paper.

---

> ### Author Response · Authors · 2024-11-21
> **New Modifications !**
>
> Thanks to all the reviewers for providing numerous valuable suggestions. After carefully checking your feedback, we have made the following revisions:
>
> 1. Provide more discussion and explanation of the design rationale behind the BNF loss in **Appendix C**.
>
> 2. Evaluate the performance of different preference optimization methods on purely mathematical datasets in **Appendix D**.
>
> 3. Present the performance of BNF under different learning rates in **Appendix E**.
>
> 4. Adjust certain phrasing to make the paper more rigorous.
>
> 5. Add a preference learning baseline, CPO.
>
> Please check the new version of this paper, Thanks!

---

> ### Author Response · Authors · 2024-11-25
> **To reviewers**
>
> Dear Reviewers,
>
> Thank you for your review! We have endeavored to address all your questions and concerns below. Please let us know if there are any aspects that we need to sufficiently clarify. If you feel that your concerns have been satisfactorily addressed, we would be grateful if you would consider revising your score. Please do not hesitate to reach out with any further questions. We value your feedback and welcome any additional queries.

---

### Official Review · Reviewer_aXkU · 2024-11-03

**Soundness:** 2
**Presentation:** 3
**Contribution:** 3
**Rating:** 6
**Confidence:** 4

**Summary:**

The paper proposes an alternative to DPO and its variants such that $|\frac{\partial{\mathcal{L}}}{\partial{z_y}}|$, the norm of the derivative of the loss with respect to the logits of a given outputs, decreases linearly as $p(y|x)$ deviates from the $p_{ref}(y|x)$ in either direction. The loss is called Bidirectional Negative Feedback, in contrast to what the paper describes as unidirectional negative feedback in the likelihood loss, namely that $|\frac{\partial{\mathcal{L}}}{\partial{z_y}}|$ increases as $p(y|x)$ decreases.

The paper also demonstrates how various DPO-series methods avoid excessive decreases in the likelihood of dispreferred samples relative to a loss that just applies NLL to the preferred samples and its negation to dispreferred samples: the gradient for these losses includes and additional scaling term $\mathcal{C}(y_w, y_l, p_\theta, p_{ref})$ that decreases as the gap between $p_\theta(y_w|x)$ and $p_{ref}(y_l|x)$ increases. However, they argue that these losses are unideal due to their sensitivity to the hyperparameter $\beta$.

Experimentally, BNF outperforms other DPO-style losses overall when evaluated across 2 instruction-following QA datasets and 4 logical reasoning datasets, using the preference training datasets from the SimPO paper. Moreover, because BNF is not a pairwise contrastive loss, it is applicable even with non-pairwise data (e.g., sometimes just a preferred or a dispreferred sample without a counterpart), and experiments show that BNF can still improve over the base model in such settings. The authors also show that BNF exhibits the least amount of log likelihood shifts as well as the lowest Gini coefficient in the logit shifts across tokens in a sequence.

**Strengths:**

1. The paper exhibits strong experimental results supporting the use of BNF over the baselines tested.
2. The paper is easy-to-follow and makes progress on a relevant topic for the ICLR community.
3. BNF is novel and well-motivated by the desire to decrease gradient norms as the log probs deviate more from the reference log probs.
4. BNF improves upon DPO-like methods empirically and in applicability (going beyond pairwise data alone).

**Weaknesses:**

1. The authors point to two problems for DPO-like baselines (i.e., training collapse and alignment tax) but only seem to show positive results of BNF for one, i.e., less alignment tax. It would be helpful to see a precise definition of training collapse and show that BNF experiences less of it (or avoid framing the paper's story with training collapse).
2. The authors mention that DPO-like baselines with an additional NLL term have limitations (i.e., poor chat and QA performance), but it is not clear to me that BNF overcomes these specific limitations when trained on the same datasets.
3. While I believe that the paper makes sound and sufficient contributions, #1 and #2 point to a misalignment between the motivation for the work in the introduction and the contributions demonstrated in the body of the paper. Section 2.3 and Figure 2 are another example; they discuss various challenges and limitations, but it is not clear how each relates back to the paper's contributions. Also, for how focused this section is on the specifics of pairwise preference data in reasoning tasks, it is strange that the experiments don't explore training on such data.
4. The paper could benefit from being more careful in distinguishing claims from hypotheses. For instance, in lines 496-471: "This suggests that BNF achieves a balanced optimization strategy, reducing the gradients for tokens already showing large differences from the reference, thereby effectively preventing over-fitting and reducing the alignment tax." The logical leaps from smaller gradients for large differences to less overfitting and less alignment tax are not directly supported in the paper. Also, in the same section, it is not clear why a lower Gini coefficient is necessarily desirable in the first place; for instance, for some sequences it may be more desirable to just decrease the probability of the incorrect tokens rather than all tokens. Moreover, lines 54-55 state: "we argue that the instability of DPO stems from a more fundamental cause: the unidirectional likelihood-derivative negative feedback inherent in log-likelihood loss." While this is a reasonable hypothesis, it is not a claim that has been definitively proven; thus, rephrasing this section as a hypothesis that guides the development of an alternative loss may be more apt.

**Questions:**

1. Could the authors provide evidence of BNF's favorability with respect to training collapse?
2. If BNF is meant to be an alternative to simply adding a NLL loss term to DPO-like methods, can the authors compare the proposed method with these baselines?
3. Could the authors explain how each point made in Section 2.3 connect to a contribution or result in the paper / positive property of BNF?
4. Could the authors clarify the central claims in the paper and distinguish them from statements that are hypotheses?
5. Could the authors explain why the BNF loss is the way it is, e.g., why it necessarily needs to look as complex as it is when the main goal is to simply avoid an increasing gradient norm for large deviations?

I would be happy to raise my score if these above concerns are addressed.

---

> ### Author Response · Authors · 2024-11-21
> **To Reviewer aXkU**
>
> Dear Reviewer,
>
> Thank you for your review! We have endeavored to address all your questions and concerns below. Please let us know if there are any aspects that we need to sufficiently clarify. If you feel that your concerns have been satisfactorily addressed, we would be grateful if you would consider revising your score. Please do not hesitate to reach out with any further questions. We value your feedback and welcome any additional queries.
>
> **Q1: Could the authors provide evidence of BNF's favorability with respect to training collapse?**
>
> Yes! We have added a section of math-related experiments in **Appendix D**, exploring the performance of different alignment methods on a purely mathematical preference dataset to further validate the effectiveness of our proposed approach.
>
> **Q2: If BNF is meant to be an alternative to simply adding a NLL loss term to DPO-like methods, can the authors compare the proposed method with these baselines?**
>
> In fact, SLiC-HF is the method with the NLL loss term. We apologize for not explicitly stating this in the paper. In the latest version, in addition to SLiC, we also included another baseline, CPO (essentially DPO+NLL), and explicitly clarified that both methods incorporate an NLL loss term.
>
> **Q3:  Could the authors explain how each point made in Section 2.3 connect to a contribution or result in the paper / positive property of BNF?**
>
> In Section 2.3, we intended to convey that the poor performance of DPO-series methods on mathematical datasets is caused by the hyper-parameter in the contrastive loss function. Specifically, the fixed hyperparameter β struggles to adapt to varying data distributions. Since our proposed BNF does not rely on contrastive loss, it can avoid this issue.
>
> We apologize for not making this logical connection clear. In the latest version of the paper, we have revised the expressions in Section 2.3 to more explicitly present this logic. Additionally, we have provided further experiments in **Appendix D**, demonstrating the performance of different preference optimization methods on pure mathematical datasets. We hope this can resolve your concerns.
>
> **Q4:  Could the authors clarify the central claims in the paper and distinguish them from statements that are hypotheses?**
>
> What you said makes a lot of sense. We have revised these inaccurate expressions in the latest version and committed to reviewing the entire paper to address similar issues. Thank you for your valuable feedback, which makes this work more rigorous.
>
> **Q5: Could the authors explain why the BNF loss is the way it is, e.g., why it necessarily needs to look as complex as it is when the main goal is to simply avoid an increasing gradient norm for large deviations?**
>
> Yes, for the first term in the BNF loss function (Equation (6)), our goal is to reduce the weight of outdated samples. When the probability of a token in the policy model has already decreased significantly compared to the reference model, further optimization becomes unnecessary.
>
> The second term in Equation (6) appears somewhat complex, but this is intentional. We structured the loss function this way to ensure the target distribution remains a valid probability distribution. Without this term, its bidirectional negative feedback properties would be affected. We have added a section in **Appendix C** to explain this in detail in the latest version of our paper.

---

> > ### Comment · Reviewer_aXkU · 2024-11-25
> > **Thanks for the response**
> >
> > Thank you to the authors for your response. My reply:
> > 1. Training collapse / Appendix D: Thanks for the additional experiment in Appendix D. If the authors could also include mention of it in the main paper, that would be helpful. Moreover, a definition of training collapse as used by the authors is still needed (i.e., degradation in performance during training), as the specific term is not used in Pal et al 2024.
> > 2. DPO + NLL baseline: Thanks, the CPO baseline was what I was looking for. It's great to see that BNF outperforms it.
> > 3. Section 2.3 and BNF: The additional text in Section 2.3 draws a more explicit connection by simply saying BNF considers a completely different loss, thanks. I would recommend removing the claim in Appendix D that the results validate the conjecture of Section 2.3, however, as it doesn't seem to me that it does. (For instance, one would ideally vary $\beta$ and see if performance improves for a contrastive loss such as DPO. I don't think such an experiment is necessary for this paper per se, but would be the type of experiment to validate the conjecture directly.)
> > 4. Distinguishing paper claims vs. hypotheses: Thanks.
> > 5. Why BNF is as complex as it is: The addition of Appendix C is useful, thanks. Though can the authors speak to why they specifically wish for the gradient norm to decrease linearly when the probability moves to either side of the reference probability, regardless of whether the direction is right or not? In particular, say a given sample is incorrect, but its probability gets updated to be larger than under the reference model (e.g., after a gradient step from a different batch of examples). Now, the model is more incorrect on this sample than before, but gradient norm with respect to the logits for this is smaller than it was before. Why is this desirable over, say, gradient norms that are bigger the further away the current probability is from the ideal?

---

> > > ### Author Response · Authors · 2024-11-25
> > > **Response to reviewer**
> > >
> > > (1) & (3) Thank you for your suggestion. We will adjust the relevant phrasing as soon as possible.
> > >
> > > (5)  Yes, the situation you described is indeed a challenging problem, and our solution is actually a trade-off. Since the supervision signal in preference optimization operates at the sentence level—where a sentence is labeled as either good or bad—we cannot simply assume that the probabilities of all tokens in non-preferred sample should be lower than before. In practice, there is often significant similarity between preferred and non-preferred samples, and the probabilities of similar tokens in non-preferred samples might increase along with those in preferred samples. Overly reducing the probabilities of these similar tokens could lead to incorrect decreases in the probabilities of positive samples. Therefore, this paper adopts a naive strategy, leaving it to the model to decide. If the probabilities of certain tokens in negative samples increase during previous learning, it can be interpreted as these tokens being "good."
> > >
> > > Of course, your suggestion is actually valuable and meaningful! That might be why PPO requires a value model to estimate the value of each token action. However, the goal of our paper is to propose a simple yet effective baseline method to replace DPO.
> > > Therefore, we choose this naive strategy.

---

> > > > ### Author Response · Authors · 2024-11-28
> > > > **Response to reviewer**
> > > >
> > > > Dear Reviewer  aXkU,
> > > >
> > > > I have revised the manuscript to address the concerns you raised, hope these revisions align with your expectations and adequately resolve your concerns. If there are any additional aspects you feel need further clarification or refinement, I would be happy to address them promptly.
> > > >
> > > > Thank you for your time and consideration, and I appreciate your willingness to reassess the manuscript in light of the updates.
> > > >
> > > > Best regards,

---

> > > > > ### Comment · Reviewer_aXkU · 2024-11-28
> > > > > **Response**
> > > > >
> > > > > Thanks for the updates. As for (5), I think the question is more about why the shape of the gradient norm graph in Figure 1 ought to be based on probability relative to the reference model rather than based on the sample's probability and identity as chosen vs. rejected. The former would seem to make it harder to correct the probability of a sample that has been moved in the wrong direction relative to the reference model, as I mentioned previously. Some discussion about this in the paper would be useful, even if just a recognition of this limitation.
> > > > >
> > > > > Regardless, I think the setting explored and evaluated in this paper is a sufficient contribution and I appreciate the authors' experimental additions and text modifications. Therefore I have raised my score.

---

> > > > > > ### Author Response · Authors · 2024-11-29
> > > > > > **To reviewer**
> > > > > >
> > > > > > We are grateful for your thoughtful feedback and for increasing your score. We are delighted that our additional experiments addressed your concerns, and your insightful comments have significantly enhanced our manuscript.
> > > > > >
> > > > > > Thank you once again for your time and consideration.

---

> ### Author Response · Authors · 2024-11-21
> **New Modifications**
>
> Thanks to all the reviewers for providing numerous valuable suggestions. After carefully checking your feedback, we have made the following revisions:
>
> 1. Provide more discussion and explanation of the design rationale behind the BNF loss in **Appendix C**.
>
> 2. Evaluate the performance of different preference optimization methods on purely mathematical datasets in **Appendix D**.
>
> 3. Present the performance of BNF under different learning rates in **Appendix E**.
>
> 4. Adjust certain phrasing to make the paper more rigorous.
>
> 5. Add a preference learning baseline, CPO.
>
> Please check the new version of this paper, Thanks!

---

> ### Author Response · Authors · 2024-11-25
> **To reviewers**
>
> Dear Reviewers,
>
> Thank you for your review! We have endeavored to address all your questions and concerns below. Please let us know if there are any aspects that we need to sufficiently clarify. If you feel that your concerns have been satisfactorily addressed, we would be grateful if you would consider revising your score. Please do not hesitate to reach out with any further questions. We value your feedback and welcome any additional queries.

---

### Author Response · Authors · 2024-11-21
**To All Reviewers**

Thanks to all the reviewers for providing numerous valuable suggestions. After carefully checking your feedback, we have made the following revisions:

1. Provide more discussion and explanation of the design rationale behind the BNF loss in **Appendix C**.

2. Evaluate the performance of different preference optimization methods on purely mathematical datasets in **Appendix D**.

3. Present the performance of BNF under different learning rates in **Appendix E**.

4. Adjust certain phrasing to make the paper more rigorous.

5. Add a preference learning baseline, CPO.

Please check the new version of this paper, Thanks!

---

### Meta-Review · Area_Chair_Zb14 · 2024-12-22

**Metareview:**

This paper was reviewed by three experts in the field and received 6, 6, 6 as the final ratings. The reviewers acknowledged that the proposed Bidirectional Negative Feedback (BNF) scheme does not require pairwise data, it has been validated through theory and extensive experiments on multiple tasks, the baseline methods are diverse and strong, and that the paper is well-structured and the ideas are clearly presented.

The reviewers raised a question about BNF's usefulness with respect to training collapse; in response, the authors have conducted experiments on a mathematical preference dataset to validate the effectiveness of their approach. A concern was raised about the complexity of the proposed BNF loss, which has been addressed by the authors in detail in the revised version of the paper. Another concern was raised about the sensitivity of BNF to hyperparameter tuning; in response, the authors have presented results depicting the performance of BNF under different learning rates, which demonstrate the stable behavior of the proposed loss function. The authors have also included a preference learning baseline method, CPO (which introduces an NLL regularization term to the DPO loss function), in response to a reviewer’s comment.

Reviewer aXkU was satisfied with the authors’ response and raised the rating from 5 to 6. During the AC-reviewer discussion period, Reviewer dQGY mentioned that he was willing to bump up his score to 7 (but couldn’t do it, as the deadline for updating scores had passed).

The reviewers, in general, have a positive opinion about the paper and its contributions. Based on the reviewers' feedback, the decision is to recommend the paper for acceptance to ICLR 2025. The reviewers have provided some valuable comments, such as the probability update scheme for incorrect samples and the statistical significance of the performance improvement of the proposed method over the baselines. The authors are encouraged to address these in the final version of their paper. We congratulate the authors on the acceptance of their paper!

**Additional Comments On Reviewer Discussion:**

Please see my comments above.

---

### Decision · Program_Chairs · 2025-01-22

Accept (Poster)